# Search-in-the-Chain: Interactively Enhancing Large Language Models with Search for Knowledge-intensive Tasks

## ABSTRACT

Making the contents generated by Large Language Model (LLM) such as ChatGPT, accurate, credible and traceable is crucial, especially in complex knowledge-intensive tasks that require multi-step reasoning and each step needs knowledge to solve. Incorporating Information Retrieval (IR) to provide LLM with external knowledge is good potential to solve this problem. However, where and how to introduce IR into LLM is a big challenge. Previous work has the problems that wrong knowledge retrieved by IR will mislead the LLM and interaction between IR and LLM breaks the reasoning chain of LLM. In this paper, we propose a novel framework named **Search-in-the-Chain** (SearChain) for the interaction between LLM and IR to solve the challenges. First, LLM generates the global reasoning chain named Chain-of-Query (CoQ) where each node consists of an IR-oriented query and the answer to the query. Second, IR verifies the answer of each node of CoQ. It corrects the answer that is not consistent with the retrieved information when IR gives high confidence, which improves the credibility. Third, LLM can indicate its missing knowledge in CoQ and rely on IR to provide this knowledge to LLM. These three operations improve the accuracy of LLM for complex knowledge-intensive tasks in terms of reasoning ability and knowledge. Finally, SearChain generates the reasoning process and marks references to supporting documents for each reasoning step, which improves traceability. Interaction with IR in SearChain forms a novel reasoning path: node-identify Depth-first Search on a tree, which enables LLM to dynamically modify the direction of reasoning. Experiments show that SearChain outperforms recent state-of-the-art baselines on complex knowledge-intensive tasks including multi-hop Q&A, slot filling, fact checking, and long-form Q&A.

## CCS CONCEPTS

• **Computing methodologies → Natural language processing**.

## KEYWORDS

Retrieval-augmented model, Large Language Models

**ACM Reference Format:**
Anonymous Author(s). 2024. Search-in-the-Chain: Interactively Enhancing Large Language Models with Search for Knowledge-intensive Tasks. In *Proceedings of the ACM Web Conference 2024 (www '24), May 13–17, 2024, Singapore.* ACM, New York, NY, USA, 17 pages. https://doi.org/10.1145/nnnnnnn.nnnnnnn

## 1 INTRODUCTION

Large Language Models (LLMs) such as ChatGPT have shown promising performance in various natural language processing tasks [2, 41]. As LLMs are gradually used in various fields, we need to pay attention to how to ensure that the contents generated by LLMs are accurate, credible and traceable, especially in the complex knowledge-intensive tasks that require multi-step reasoning and each step needs corresponding knowledge to solve [20, 37]. Many studies have shown that LLMs have trouble in: (1) compositional reasoning over multiple knowledge [21], (2) memorization of long-tail and real-time knowledge [12] and (3) avoiding hallucination that is inconsistent with the facts [1], which affects the accuracy and credibility of LLMs for complex knowledge-intensive tasks. Besides, context-only generation without any supporting evidence causes less traceability and makes people less trust in the LLM-generated content. Retrieval-augmented method has good potential to solve these problems because it combines the knowledge of the model with external knowledge bases [9, 11, 17].

However, how to introduce IR into LLM is not a trivial thing. There are three main challenges. *C*-1: Directly inserting IR into the reasoning process of LLM such as Self-Ask [21], LTM [42], React [36] and DSP [14] leads to breaking the reasoning chain of LLM. Because in these methods, LLM can only reason a local sub-question in each generation. Although AgentGPT and PS [31] first plan sub-questions and then solve them, they are not suitable for scenarios where the next sub-question is dependent on the answer of the previous sub-questions, which is common for complex tasks, such as the multi-hop QA. *C*-2: When there is a conflict in the knowledge of IR and LLM, for the knowledge that the LLM has correctly memorized, it risks being misled by IR if IR retrieves the wrong information. Therefore, it is important to decouple the knowledge of LLM and IR to make sure that IR only provides the knowledge that LLM really needs to avoid the misleading of LLM by IR, which is not considered in previous methods. *C*-3: Previous methods cannot dynamically modify the reasoning direction.

In this paper, we propose a novel framework named Search-in-the-Chain (SearChain) to effectively combine LLM with IR to solve the above challenges (Figure 1). SearChain and previous methods both need multiple IR-LLM interaction rounds, but the former works at the chain level, while the latter only deals with a node. In each round, SearChain performs reasoning, verification, and completion. After the interaction, SearChain performs tracing to generate the final content. Specifically, in each round, first, LLM exploits in-context learning to construct a Chain-of-Query (CoQ), which is a reasoning chain to decompose and solve complex questions. Each node of the chain consists of an IR-oriented query, the answer generated by LLM for this query and a flag indicating whether LLM needs additional knowledge. Different from previous methods in which LLM can only perform one-step reasoning (only a node) when interacting with IR, CoQ is a complete chain. This design

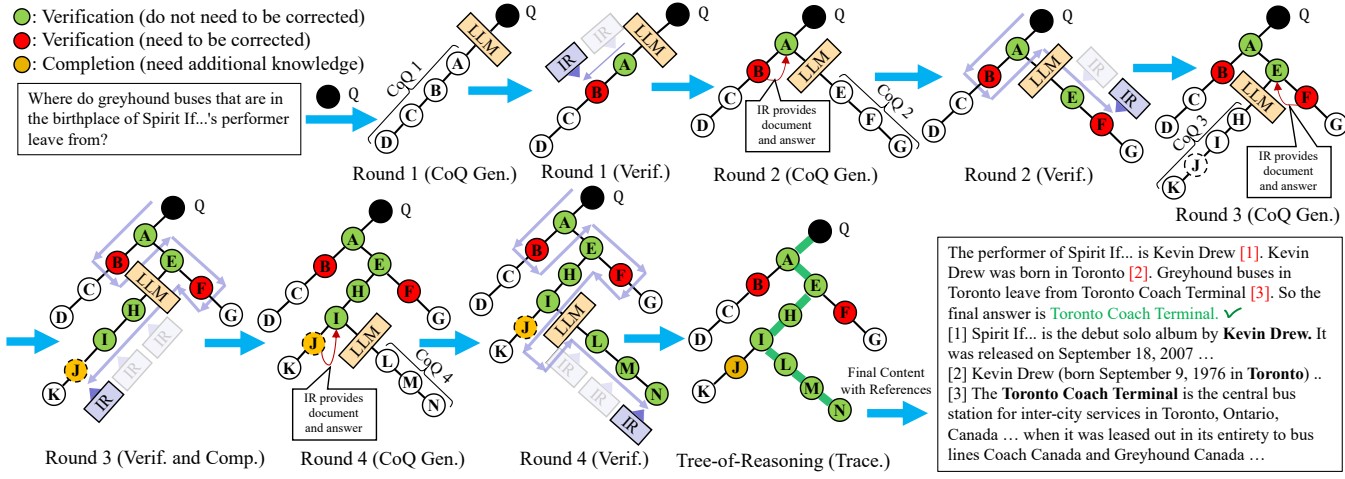

**Figure 1: Interaction between IR and LLM in SearChain. First, SearChain makes LLM plan a CoQ where each node is a query-answer pair. Then, IR interacts with each node of CoQ to perform verification and completion. If IR detects that a node needs to be corrected or provided with knowledge, it gives feedback to LLM and LLM re-generates a new CoQ, which is the new branch of the tree. This process is the node-identify Depth-first Search on a tree called Tree-of-Reasoning (correct reasoning path is green). Final content includes the reasoning process and references to supporting documents.**

avoids IR from breaking the reasoning chain (**C-1**). Second, IR interacts with each node of CoQ to perform verification and completion. In verification, IR verifies the answer of each node. In case when the LLM-generated answer is not consistent with the retrieved information and IR gives high confidence, IR gives feedback to LLM to help it correct the answer and re-generate the correct CoQ. In completion, IR determines whether the node has missing knowledge from the flag of the node and provides this knowledge to LLM to help it re-generate CoQ. LLM gradually generates the correct CoQ through multiple rounds of interaction with IR. The above design provides LLM with the knowledge it really needs to alleviate the misleading caused by IR to LLM (**C-2**), which improves accuracy. IR verifies and corrects the knowledge in the reasoning process of LLM based on external knowledge bases, which improves credibility. After the interaction, SearChain performs tracing to generate the reasoning process and marks references to supporting documents for each reasoning step, which is used as the final content returned to the user. This improves the traceability of knowledge in the generated contents. Interaction with IR in SearChain transforms the reasoning path from a chain to node-identify Depth-first Search on a tree called Tree-of-Reasoning (ToR). CoQ generation can be seen as a part of Depth-first Search and IR can identify the nodes that need more information (**C-3**). This enables LLM to dynamically modify the reasoning direction. This paper's main contributions are:

**(1)** We highlight the challenges in introducing IR into LLM from the perspectives of reasoning and knowledge.

**(2)** We propose a novel framework called SearChain to effectively combine LLM and IR. SearChain not only stimulates the knowledge-reasoning ability of LLM but also uses IR to identify and offer the knowledge that LLM really needs, which helps to improve accuracy and credibility. Moreover, SearChain can also mark references to supporting documents for the knowledge involved in the generated contents, which helps to improve the traceability of the knowledge.

**(3)** Interaction with IR in SearChain forms a novel reasoning path: node-identify Depth-first Search on a tree, which enables LLM to dynamically modify the direction of reasoning.

**(4)** Experiment shows that SearChain outperforms state-of-the-art baselines on complex knowledge-intensive tasks including multi-hop question-answering, slot filling, fact checking and long-form question-answering. Code will be released at https://github.com.

## 2 RELATED WORK

### 2.1 Chain-of-Thought Prompting

Chain-of-thought [33] proposes the method that uses few-shot examples to enable LLM to give intermediate reasoning results when solving complex problems and improves the reasoning ability. [15] uses "Let's do it step by step" as prompt to achieve promising zero-shot performance. Auto-CoT exploits language models to automatically construct few-shot learning examples for CoT [39]. There are also many studies that cover other aspects of CoT such as self-consistency [32], usage of small and medium size models [38] and selection [7]. Besides, there are studies that iteratively use LLM to decompose complex questions and answer sub-questions step by step. These methods include Least-to-Most [42], Dynamic Least-to-Most [4], Self-Ask [21] and DSP [14]. Chain-of-Query of our method is also inspired by CoT. However, previous studies focus on giving intermediate reasoning results or decomposing complex questions and answering sub-questions step by step. They focus on how to solve local sub-questions while ignoring the global planning of the reasoning chain. Although AgentGPT and PS [31] first plan each sub-question and then solve them, they are not suitable for scenarios where the next sub-question needs the answer of the previous sub-questions to generate, which is common for complex knowledge-intensive tasks (multi-hop QA). CoQ of our method makes LLM construct a global reasoning chain where each node is

a query-answer pair. This design not only improves the knowledge-reasoning ability but also provides the interface for IR to be deeply involved in the reasoning process of LLM.

## 2.2 Retrieval-augmented Language Models

Many studies have shown that retrieval-augmented methods can connect language models with external corpus to get promising performance in various natural language tasks such as open-domain question answering [9, 11, 17], language modeling [3, 19] and enhancing the factuality [22]. Recently, some studies enable LLM to interact with IR via in-context learning [14, 21, 25, 36]. In these methods, the interaction between IR and LLM makes the reasoning of LLM not continuous. LLM can only perform one-step reasoning at each inference. Our method makes LLM generate a global reasoning chain called Chain-of-Query at each inference, the hidden states in generation are used as the sequential dependency, which introduces stronger logical relationship between each reasoning step. Besides, previous methods can only provide information to the LLM but cannot assist LLM in correcting erroneous information or avoid the negative effect of IR on LLM, which makes the reasoning of LLM still in a one-dimensional chain. Our method makes IR interact with each node of the chain. IR only provides LLM with its missing knowledge and corrects the answers that are not consistent with the retrieved information when IR is confident enough. This design mitigates the negative effect of IR on LLM and transforms the reasoning path from chain to node-identify Depth First Search on a tree to enable LLM to modify the reasoning direction.

## 3 OUR METHOD

This section introduces the design of Search-in-the-Chain (SearChain). In SearChain, IR and LLM conduct multiple rounds of interaction. In each round, first, LLM acts as the commander to plan the global reasoning chain for the complex input questions called Chain-of-Query (CoQ). Each node of the CoQ consists of an IR-oriented query, the answer generated by LLM for this query and a flag indicating whether LLM needs additional knowledge. Then, IR interacts with each node of CoQ and performs the completion and verification to provide LLM with missing knowledge and correct the wrong answers. LLM re-generates new CoQ based on feedback from IR. Multiple rounds of interaction help LLM to gradually generate the correct CoQ according to the external knowledge base, which improves accuracy and credibility. Finally, SearChain performs tracing to generate the whole reasoning process and marks references to supporting documents for each reasoning step, which is used as the final content returned to the user. This improves the traceability of generated content. Interaction with IR in SearChain transforms the reasoning path from a chain to node-identify Depth-first Search on a tree called Tree-of-Reasoning (ToR), which enables LLM to dynamically modify the reasoning direction. Besides, SearChain decouples the knowledge of LLM and IR and provides LLM with the knowledge it really needs to alleviate the misleading of LLM.

## 3.1 Comparison with Previous Methods

Figure 2 shows the difference between our method and previous retrieval-augmented methods (Self-Ask [21], React [36], DSP [14], etc.) in solving complex knowledge-intensive questions.

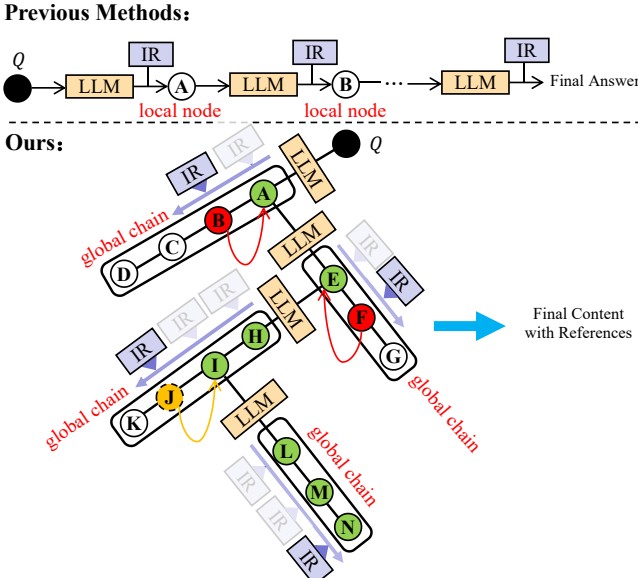

**Figure 2: Comparison with previous methods.**

**(1) Local vs. Global.** For a complex question that needs multi-step reasoning, previous methods directly insert IR into the multi-step reasoning process, causing LLM can only reason a **local** sub-question such as node Ⓐ in each generation. This breaks the reasoning chain of LLM. Our method proposes Chain-of-Query to provide the interactive interface for IR on the premise of ensuring the coherence of reasoning chain (plan a **global** chain for question $Q$ such as Ⓐ→Ⓑ→Ⓒ→Ⓓ in each generation). (solves *C*-1)
**(2) Directly Provide vs. Verify and Complete.** Previous methods directly provide the retrieved information to the LLM. When the retrieved information is incorrect, the LLM runs the risk of being misled. In our method, IR only corrects inconsistent information in Chain-of-Query when IR is confident enough, and provides the information that LLM does not know via flags on Chain-of-Query, which mitigates the negative effect of IR on LLM. (solves *C*-2)
**(3) Chain vs. Tree.** Previous methods cannot modify the reasoning direction in time as necessary. Our method transforms the reasoning path from a chain to node-identify Depth-first Search on a tree by introducing the verification and completion from IR, which enables LLM to dynamically modify the direction of reasoning. (solves *C*-3)

## 3.2 Chain-of-Query Generation

In SearChain, we use in-context learning [33] to prompt large language model to construct a global reasoning chain for complex question $Q$ named Chain-of-Query (CoQ):

$$\text{CoQ} = (q_1, a_1) \rightarrow (q_2, a_2) \rightarrow ... \rightarrow (q_n, a_n), \tag{1}$$

which is the branch of Tree-of-Reasoning. Each node $(q_i, a_i)$ of CoQ consists of an IR-oriented query $q_i$ and its answer $a_i$. $q_1 ... q_n$ are the sub-questions that need to be solved in the reasoning process of solving $Q$. CoQ generation is applied to each round of interaction between LLM and IR. In the first round, the prompt used to make LLM generate CoQ is shown in Figure 3. The prompt

Figure 3: Prompt to make LLM generate Chain-of-Query.

starts with "*Construct a global reasoning chain*" to make LLM know that the main task is to generate a global reasoning chain in each generation. **"Global"** means that LLM needs to plan a complete reasoning chain for the complex question, rather than answer the question directly or only solve **"local"** sub-questions (comparison shown in Figure 2). At each node of the chain, LLM focuses on generating the IR-oriented query and gives the answer if LLM knows. If LLM does not know the answer, it should mark the query with "[Unsolved Query]", which is a flag indicating the missing of knowledge. In subsequent rounds, when a node needs IR to correct or provide missing knowledge, LLM generates a new CoQ according to the feedback of IR to dynamically modify the reasoning direction. The design for this scenario will be introduced in Section 3.3. The generation of CoQ is a complete Depth-first Search for $Q$, which avoids IR from breaking the reasoning chain of LLM. Experiments (Section 4.3.3) also show that for the difficult sub-question, CoQ enables LLM to solve it by more reasoning steps such as rewriting or further decomposing the sub-question while baselines tend to stop reasoning. It is because baselines focus on solving current local sub-questions while ignoring the global planning of the reasoning chain. The global perspective in CoQ makes LLM try harder to explore possible answers when facing intermediate difficulties.

## 3.3 Interaction with Information Retrieval

In each round of interaction, LLM passes the generated CoQ to IR. IR verifies and completes the information for each node $(q_i, a_i)$ of CoQ and feeds back to LLM to help it generate more correct CoQ as the new branch of ToR (Tree-of-Reasoning). Besides, IR records the corresponding retrieved documents for each node of

---

**Algorithm 1:** Description of the Interaction with IR.

**Initialize :** Processed queries: $M = null$;
        Correct reasoning path: $R = null$;
        Interaction rounds: $r = 0$;
        Feedback: $F = null$; ToR: $T = Q$;

**Function** $\mathtt{IR}(q_i, a_i)$:
  $d_i = \mathtt{Retrieval}(q_i)$; // *Retrieve Top-1 document $d_i$ for $q_i$.*
  $g, f = \mathtt{Reader}(q_i, d_i)$;
  // *Extract answer $g$ from $d_i$ and give confidence $f$.*
  **if** $q_i$ is Unsolved Query **then**
    // *Completion.*
    $R$.add $(q_i, g, d_i)$;   // *Record the correct node.*
    **return** $\mathtt{PromptForComplete}(q_i, g, d_i)$;
  **if** $f > \theta$ and $\mathtt{NotEqual}(g, a_i)$ **then**
    // *Verification.*
    $R$.add$(q_i, g, d_i)$;   // *Record the correct node.*
    **return** $\mathtt{PromptForVerify}(q_i, g, d_i)$;
  $R$.add $(q_i, a_i, d_i)$; **return** "Pass" ;

**Function** $\mathtt{Traverse}(CoQ)$:
  **foreach** $(q_i, a_i)$ in $CoQ$ **do**
    **if** not $\mathtt{DuplicateQuery}(q_i, M)$ **then**
      // *If $q_i$ has not been processed.*
      $F = \mathtt{IR}(q_i, a_i)$; $M$.add$(q_i)$;
      **if** not F == "*Pass*" **then return** F ;
  **return** "*Finish*" ;

**Function** $\mathtt{Main}(Q, F)$:
  **while** not (F == "Finish" or $r > r_{max}$) **do**
    $CoQ = \mathtt{ChainGenerate}(Q, F)$;
    // *LLM generate the new Chain-of-Query CoQ.*
    $T$.AddChild($CoQ$); // *Add the new branch to $T$.*
    $F = \mathtt{Traverse}(CoQ)$; // *Interact with IR.*
    $r = r + 1$; // *Update the number of interaction rounds $r$.*
  **return** $\mathtt{Tracing}(T, R)$

---

CoQ as its supporting documents, which enhances the traceability of LLM-generated content. The description of interaction is shown in Algorithm 1. IR interacts with each node $(q_i, a_i)$ of CoQ, retrieves the Top-1 document $d_i$ for $q_i$ as the supporting document, and judges whether to verify or complete it according to the type of $q_i$. When all the queries of CoQ do not need to be corrected or completed, or the maximum number of interaction rounds is reached, the interaction ends. SearChain traces back the correct reasoning path of ToR and refers to each node of the path to generate the final content with marked references to supporting documents for knowledge of each node.

**Verification.** Verification aims to guarantee the correctness of $a_i$ in each node $(q_i, a_i)$ of CoQ based on the external knowledge base, which improves the accuracy and credibility of generated content. Specifically, given the retrieved Top-1 document $d_i$ for $q_i$, a Reader [13] that has been trained on open-domain QA datasets [13] is used to extract the answer $g$ for $q_i$ from $d_i$ with its confidence $f$ ($f$ is a predicted value that measures whether $g$ can answer $q_i$):

$$s = \arg\max(\mathrm{softmax}(\mathbf{H}\mathbf{w}_s)), e = \arg\max(\mathrm{softmax}(\mathbf{H}\mathbf{w}_e)),$$

$$g = d_i[s:e], f = \mathbf{H}_{[CLS]}\mathbf{w}_f, (\mathbf{w}_s, \mathbf{w}_t, \mathbf{w}_f \in \mathbb{R}^E),$$

where $\mathbf{H} \in \mathbb{R}^{L \times E}$ is the sequence of last hidden states for the input text "[CLS]$q_i$[SEP]$d_i$", $L$ is the length and $E$ is hidden dimension. $\mathbf{H}_{[CLS]}$ is the last hidden state of [CLS] token. Then, SearChain judges whether the answer $a_i$ given by LLM is consistent with the retrieved information according to (1) whether $g$ appears in $a_i$ (for short-form generation tasks such as multi-hop QA and slot filling) or (2) whether ROUGE [18] between $a_i$ and $d_i$ is greater than the threshold $\alpha$ (for long and free-form generation tasks such as ELI5 [6]). If $a_i$ is not consistent with retrieved information and Reader is confident enough ($f > \theta$, $\theta$ is a threshold to alleviate the negative effect of IR on LLM), a prompt is constructed to help LLM correct the answer $a_i$. The template of the prompt is: *"According to the Reference, the answer for $q_i$ should be g, you can change your answer and continue constructing the reasoning chain for [Question]: Q. Reference: $d_i$."*. This round is over. LLM receives the feedback of IR, gives the new answer $a_i'$ for $q$, and generates a new CoQ with $(q_i, a_i')$ as the root node, which is the new branch of ToR.

**Completion.** Completion aims to provide LLM with missing knowledge in nodes of CoQ, which improves the accuracy of generated contents. Specifically, in CoQ generation (Section 3.2), LLM marks "[Unsolved Query]" for the unsolvable query. For the unsolvable query $q_i^*$, IR extracts the answer $g^*$ from retrieved document $d_i^*$ as described in Verification. Regardless of whether $f$ is greater than the threshold $\theta$, $g^*$ and $d_i^*$ will be fed back to the LLM in the form of a prompt because the LLM cannot solve $q_i^*$. The template of the prompt is: *"According to the Reference, the answer for $q_i^*$ should be $g^*$, you can give your answer and continue constructing the reasoning chain for [Question]: Q. Reference: $d_i^*$."*. This round is over. LLM receives the feedback, gives the answer $a_i^*$ to solve the query $q_i^*$ and generates a new CoQ with $(q_i^*, a_i^*)$ as the root node, which is the new branch of ToR.

**Tracing.** Tracing aims to generate the reasoning process and mark references to supporting documents for each reasoning step, which is used as the final content returned to the user. This improves the traceability of each knowledge in the generated content. Specifically, SearChain records the documents retrieved for each node on the correct reasoning path of Tree-of-Reasoning as the supporting documents. SearChain prompts LLM to generate the final content by referring to nodes on the correct path and mark references to the supporting documents for the corresponding sub-fragments of the generated content (final content of Figure 1). The prompt is *"You can try to generate the final answer for the [Question] by referring to the [Query]-[Answer] pairs, starting with [Final Content]. [Query 1]: $q_1$. [Answer 1]: $a_1$ ... [Query m]: $q_m$. [Answer m]: $a_m$."*. This design enables the user to acquire the related documents of the knowledge involved in each step of reasoning. We believe that it is a promising task to mark references to supporting documents on sub-fragments of complex content generated by LLM. Our approach provides a novel and effective approach to solve this task by retrieving supporting documents for each sub-questions involved in the reasoning process of LLM without any supervised data (texts with citation annotations) and training of the LLM.

**Node-identify Depth-first Search.** Compared with previous retrieval-augmented methods, interaction with IR in SearChain forms a novel reasoning path: node-identify Depth-first Search

on a tree. In each generation, LLM generates a CoQ to perform continuous reasoning on complex questions until the final answer is generated or an unsolvable sub-question is encountered. This can be seen as a part of Depth-first Search (DFS). However, different from traditional DFS algorithm [28], "node-identify" in SearChain means that when a search in one direction is terminated, SearChain does not return to its parent node, but dynamically identifies the node that needs to be corrected or completed via verification and completion in IR and re-generates a new CoQ started with this node. The interaction process between IR and LLM in SearChain is the process of constructing a tree using node-identify DFS, which enables LLM to dynamically modify the reasoning direction.

## 4 EXPERIMENTS

In this section, we compare SearChain with recent related baselines on complex knowledge-intensive tasks and conduct the analysis.

### 4.1 Experimental Setup

*4.1.1 **Datasets and Evaluation Metric**.* We select four classic complex knowledge-intensive tasks including multi-hop question-answering (HotpotQA (HoPo) [34], Musique (MQ) [30], WikiMulti-HopQA (WQA) [10] and StrategyQA (SQA) [8]), slot filling (zsRE [16], T-REx [5]), fact checking (FEVER [29]) and long-form question-answering (ELI5 [6]). These tasks require LLM to perform multi-step reasoning on complex questions, and each step requires corresponding knowledge to solve. As for the evaluation metrics, for ELI5 whose ground truth is long and free-form, we use ROUGE-L [18] as the metric. For other tasks, we use whether the ground truth answer is contained within the generated answer (i.e, cover-EM [23]) as the metric. Following DSP [14] and Self-Ask [21], we evaluate the model on full development datasets of MQ and HoPo, BIG-bench [26] datasets on SQA and subsets of WQA, zsRE, T-REx, FEVER and ELI5 (each subset has 1.2k questions).

*4.1.2 **Baselines**.* Our baselines can be divided into two categories, one is about improving the reasoning ability of LLM on complex tasks (CoT [33], CoT-SC [32], Auto-CoT [39], Recite-and-answer [27] and Least-to-Most [42]), and the other is not only introducing IR to LLM but also improving the reasoning ability (Direct[1], Self-Ask [21], ToolFormer[2] [25], React [36], DSP [14], Verify-and-Edit (combined with CoT-SC) [40] and Tree-of-Thought [35]). AgentGPT and PS [31] use Plan-and-Solve paradigm, we also reproduce this as one of the baselines.

*4.1.3 **Implementation**.* The large language model we used is *gpt-3.5-turbo* provided from API of OpenAI[3] and the retrieval model we used is ColBERTv2 [24] (following DSP). IR model infers on one Tesla V100 GPU. For HotpotQA, we use Wikipedia 2017 as the corpus, which is provided by [34] in full-wiki setting. For the other datasets, we use the large-scale passage collection built on Wikipedia as the corpus [13]. Baselines with information retrieval are in the same setting as SearChain. We reproduce all baselines on *gpt-3.5-turbo* following the settings in their papers. The maximum number of interaction rounds $r_{max}$ is 5. The thresholds $\alpha$ and $\theta$ are

---

[1]Retrieve documents and provide them to LLM in a prompt.
[2]Perform ToolFormer on *gpt-3.5-turbo* via in-context learning.
[3]https://openai.com/api/

**Table 1: Performance of SearChain and baselines on complex knowledge-intensive tasks. Bold denotes the best result in different settings. FC: Fact Checking, LFQA: Long-Form QA. Metric for LFQA: ROUGE-L. Metric for others: cover-EM.**

| | Muti-Hop QA | | | | Slot Filling | | FC | LFQA |
|---|---|---|---|---|---|---|---|---|
| | HoPo | MQ | WQA | SQA | zsRE | T-REx | FEV. | ELI5 |
| *Without Information Retrieval* | | | | | | | | |
| Direct Prompting | 31.95 | 5.91 | 25.82 | 66.25 | 22.75 | 43.85 | 73.45 | 21.90 |
| Auto-CoT | 33.53 | 10.55 | 29.15 | 65.40 | 21.30 | 43.98 | 76.61 | 21.55 |
| CoT | 35.04 | 9.46 | 30.41 | 65.83 | 22.36 | 44.51 | 76.98 | 21.79 |
| CoT-SC | 36.85 | 10.02 | 32.68 | 70.84 | 24.74 | 46.06 | 77.15 | 22.05 |
| Recite-and-answer | 36.49 | 10.97 | 32.53 | 70.47 | 24.98 | 46.14 | **77.35** | 22.10 |
| Self-Ask w/o IR | 33.95 | 11.10 | 35.65 | 65.45 | 20.16 | 44.71 | 75.31 | 21.73 |
| Least-to-Most | 34.05 | 11.45 | 32.88 | 65.78 | 21.86 | 44.98 | 75.98 | 21.95 |
| Plan-and-Solve | 36.33 | 12.95 | 35.68 | 73.21 | 25.15 | 47.58 | 77.08 | 22.23 |
| SearChain w/o IR | **38.36** | **13.61** | **40.49** | **75.62** | **30.14** | **52.69** | 77.06 | **22.54** |
| *Interaction with Information Retrieval* | | | | | | | | |
| Direct Retrieval | 34.09 | 10.22 | 30.01 | 66.78 | 52.29 | 59.28 | 78.25 | 23.40 |
| ToolFormer | 36.75 | 12.98 | 35.49 | 67.02 | 51.35 | 59.17 | 80.79 | 23.05 |
| Self-Ask | 40.05 | 14.28 | 39.58 | 67.65 | 50.51 | 59.12 | 79.41 | 23.25 |
| Plan-and-Solve w/ IR | 41.65 | 15.07 | 42.05 | 74.58 | 52.15 | 60.03 | 81.04 | 24.56 |
| React → CoT-SC | 43.15 | 15.49 | 40.36 | 70.43 | 53.27 | 60.42 | 80.59 | 24.05 |
| Verify-and-Edit | 44.03 | 15.57 | 40.83 | 71.09 | 53.95 | 61.10 | 80.67 | 23.80 |
| Tree-of-Thought w/ IR | 50.65 | 15.61 | 42.49 | 72.55 | 54.88 | 62.40 | 81.03 | 24.20 |
| DSP | 51.97 | 15.83 | 43.52 | 72.41 | 54.35 | 61.32 | 80.65 | 23.46 |
| SearChain | **56.91** | **17.07** | **46.27** | **76.95** | **57.29** | **65.07** | **81.15** | **25.57** |
| - w/o Verification | 46.11 | 14.70 | 42.67 | 75.98 | 43.58 | 55.46 | 78.79 | 22.98 |
| - w/o Completion | 53.05 | 15.86 | 43.64 | 76.53 | 45.78 | 56.03 | 80.03 | 25.02 |

set as 0.35 and 1.5 respectively. As for the selection of confidence threshold ($\theta$), we initialize the initial value of the confidence threshold (1.0) based on prior knowledge and gradually increase the value with a step size of 0.1. We validate the F1-score (a comprehensive metric of the Recall and Precision of judging whether the passage can answer the question) on the mixed open-domain QA datasets (NQ, TriviaQA, WebQ, and TREC) after each value change. We find that when the confidence threshold is 1.5, the highest F1-score can be achieved so we set the confidence threshold as 1.5. As for the selection of ROUGE threshold ($\alpha$), we determine this value by observing the ROUGE relationship between the generated text and the ground truth in the few examples in in-context learning. Our further experiments show that when the value range of ROUGE threshold is between 0.3 and 0.5, the performance change on ELI5 is not obvious. Details of prompts and experiments are introduced in Section A.4 of Appendix.

## 4.2 Main Results

Performance of SearChain and baselines on complex knowledge-intensive tasks are shown in Table 1.

**(1) Effect of Chain-of-Query.** CoQ is the reasoning chain for complex questions in SearChain. We compare it with recent competitive baselines in the setting without IR. SearChain w/o IR outperforms all baselines based on CoT (CoT, Auto-CoT, CoT-SC and Recite-and-answer), which indicates that focusing on constructing a global reasoning chain consisting of sub-questions is better than just giving intermediate reasoning results. SearChain w/o IR

outperforms Self-Ask w/o IR and Least-to-Most, which indicates that it is more effective to focus on constructing a global reasoning chain at each inference (global perspective) than generating and answering sub-questions step by step (local perspective).

**(2) Effect of interaction with IR.** In the setting with interaction with IR, SearChain again outperforms all the baselines. The paradigm of first generating global CoQ, and then IR interacting with each node of CoQ ensures the coherence of LLM reasoning. This solves the problem in Self-Ask, DSP and React. Besides, SearChain decouples the knowledge of LLM and IR. IR judges whether to provide information to LLM according to the confidence and the flag of the node on CoQ, which effectively alleviates misleading LLM. Last but not least, baselines reason in the one-dimensional chain. They cannot dynamically modify the reasoning direction. Interaction with IR in SearChain transforms the reasoning path from a chain to node-identify Depth-first Search on a tree, which enables LLM dynamically modify the reasoning direction.

## 4.3 Analysis

In this section, we discuss and compare the advantages of SearChain compared to baseline in detail. First, we analyze the source of the knowledge of SearChain in solving complex questions. Second, while we analyze the positive effect of IR on LLM in solving difficult questions, we also demonstrate that our method can better mitigate the negative effect of IR on LLM. Third, we show the advantages of SearChain compared to baselines in terms of reasoning and tracing capabilities. Last but not least, we perform efficiency analysis to

**Table 2: Distribution of knowledge sources.**

| Knowledge Src. | HoPo | MQ | WQA | SQA |
|---|---|---|---|---|
| LLM | 74.56% | 78.83% | 75.83% | 94.98% |
| Corrected by IR | 20.94% | 14.60% | 18.96% | 2.78% |
| Completed by IR | 4.50% | 6.57% | 5.21% | 2.24% |

**Table 3: Positive and negative effects of IR on LLM.**

(a) Accuracy on $\mathbb{S}_{IR}$ and $\mathbb{S}$ (positive effect ↑).

| | HoPo | MQ | WQA | SQA |
|---|---|---|---|---|
| w/o IR ($\mathbb{S}$) | 38.36 | 13.61 | 40.49 | 75.62 |
| w/o IR ($\mathbb{S}_{IR}$) | 31.38 | 10.20 | 32.60 | 68.96 |
| w IR ($\mathbb{S}_{IR}$) | 60.86 | 18.49 | 50.52 | 78.42 |

(b) Percentage that IR misleads LLM (negative effect ↓).

| | HoPo | MQ | WQA | SQA |
|---|---|---|---|---|
| Self-Ask | 15.76 | 14.32 | 25.76 | 10.29 |
| React | 17.68 | 15.22 | 25.99 | 10.03 |
| Plan-and-Solve w/ IR | 16.42 | 15.25 | 22.31 | 7.59 |
| Verify-and-Edit | 9.78 | 10.75 | 16.44 | 6.52 |
| Tree-of-Thought w/ IR | 12.07 | 13.25 | 20.52 | 8.46 |
| DSP | 14.72 | 14.03 | 24.31 | 9.22 |
| SearChain | 6.33 | 6.50 | 12.71 | 5.31 |

show our method significantly improves task performance with no significant increase in time consumption.

*4.3.1 **Knowledge Decoupling**.* We analyze the knowledge sources on the four multi-hop QA datasets. Specifically, we classify knowledge sources into three categories: (1) knowledge of LLM, (2) knowledge that corrected by IR in verification, and (3) knowledge that LLM does not know and is provided by IR in completion. We use node of ToR as the statistical granularity to calculate the percentage of nodes from these three sources in the total nodes respectively. The experimental results are shown in Table 2. It is worth noting that even though most of the knowledge comes from LLM, this knowledge is also verified by IR. IR only corrects the inconsistent answer given by LLM when it is confident enough and provides LLM with the missing knowledge, which alleviates the negative effect of IR on LLM and improves utilization of retrieved information. On StrategyQA, LLM has memorized most knowledge that IR can retrieve, so IR provides less knowledge than other datasets.

*4.3.2 **Positive and Negative Effects of IR on LLM**.* **(1) Positive.** In SearChain, IR can identify the trouble of LLM and effectively help LLM to correct the answers and acquire missing knowledge. We select the questions ($\mathbb{S}_{IR}$) that IR helps to correct or provide knowledge from the datasets used in Table 1 ($\mathbb{S}$) and evaluate the accuracy of SearChain on $\mathbb{S}_{IR}$. We also evaluate the accuracy of SearChain w/o IR on $\mathbb{S}_{IR}$. The results in Table 3(a) show that w/o IR performs worse on $\mathbb{S}_{IR}$ than on $\mathbb{S}$, which indicates that LLM does have trouble with the questions that require IR help. w/ IR

**Table 4: Number of reasoning steps. SearChain tries more for unsolvable sub-questions to achieve better accuracy.**

| | 2-hop | 3-hop | 4-hop | Accuracy |
|---|---|---|---|---|
| CoT | 2.25 | 2.23 | 2.19 | 35.04 |
| Self-Ask w/o IR | 2.04 | 2.21 | 2.15 | 33.95 |
| Least-to-Most | 2.52 | 2.68 | 2.70 | 34.05 |
| SearChain w/o IR | 4.16 | 4.66 | 5.06 | 38.36 |

**Figure 4: Case study of the difference between SearChain and baselines for unsolvable sub-questions.**

performs better on $\mathbb{S}_{IR}$, which indicates that IR effectively identifies and solves the trouble of LLM.

**(2) Negative.** We point out the risk of IR misleading LLM when there is a conflict in the knowledge of IR and LLM (Section 1). We select the questions ($\mathbb{S}_t$) that LLM can give the correct answers to and count the percentage that LLM gives incorrect answers after adding IR on $\mathbb{S}_t$. Table 3(b) shows SearChain effectively mitigates the negative effect of IR on LLM. It is because that SearChain uses the confidence of IR and the information of CoQ to judge whether to correct LLM or provide LLM with its missing knowledge.

*4.3.3 **CoQ vs Baselines in Reasoning**.* CoQ performs better on reasoning complex questions than the baselines. In addition to Table 1, we further analyze the reasoning ability from two aspects:

**(1) Number of Reasoning Steps.** We analyze the number of reasoning steps in different methods in the setting without IR. We conduct the experiment on Musique because Musique has more complex questions. Table 4 shows the average number of reasoning steps on questions with different hops. Our method has more reasoning steps, and the number of reasoning steps increases with the hops of the question. This shows that our method has a better perception of the complexity of the questions.

**(2) Solving Difficult Sub-questions.** The baselines focus on solving local sub-questions while ignoring the global planning of the reasoning chain. This leads LLM to tend to stop reasoning rather than try more when a sub-question cannot be solved. In our method, LLM acts as a commander that plans a global reasoning chain that can solve the complex question, when a sub-question cannot be solved, even without the help of IR, LLM can try to further decompose or rewrite the sub-question to continue reasoning. It is because our method focuses on building a global chain that can solve the

| 🌿 SearChain | 🔷 New Bing | 🌿 SearChain | 🔷 New Bing | 🌿 SearChain | 🔷 New Bing |
|---|---|---|---|---|---|
| **What team did the recipient of the 2007 Brownlow Medal play for?** | | **What kind of dinosaur does Raymond Ochoa voice in the Pixar film "The Good Dinosaur"?** | | **In what London botanical garden can you find the fungus described by Elise Maud Wakefield as "Lepiota"?** | |
| Jimmy Bartel was the recipient of the 2007 Brown-low Medal [1] and played Australian rules football professionally [2]. He played for the Geelong Football Club in the Australian Football League [2]. So the final answer is Geelong Football Club.

[1]: … Jimmy Bartel won the medal during the 2007 AFL season …
[2]: … Jimmy Bartel is a former Australian rules footballer plays for the Geelong Football Club … | The recipient of the 2007 Brownlow Medal was Jimmy Bartel who played for the Geelong Football Club. He won the medal by polling twenty-nine votes during the 2007 AFL season [1]. Is there anything else you would like to know about this topic?

[1]: … Jimmy Bartel won the medal by polling twenty-nine votes during the 2007 AFL season… | Raymond Ochoa is an American child actor and voice actor [1]. He has played various roles in movies including Arlo in "The Good Dinosaur" [2]. Arlo is an Apatosaurus [3]. So the final answer is Apatosaurus.

[1]: Raymond Ochoa is an American child actor and voice actor …
[2]:Raymond Ochoa … played, Arlo, in the Pixar film "The Good Dinosaur" …
[3] … timid Apatosaurus named Arlo (Ochoa) … | Raymond Ochoa voices the character of Arlo, an Apatosaurus in the Pixar film "The Good Dinosaur" [1]

[1]:(Amazon.com: The Good Dinosaur : Raymond Ochoa) | Elise Maud Wakefield was an English mycologist and plant pathologist [1]. "Lepiota" is a genus of fungi [2]. Kew Gardens is known for its fungi collection [3], such as "Lepiota viridigleba" [4]. So the final answer is Kew Gardens.

[1]: … Elsie Maud Wakefield, was an English mycologist and plant pathologist…
[2]: … Lepiota viridigleba is a species of sequestrate fungus in the family Agaricaceae …
[3] … Kew Gardens is a botanical garden in southwest London that houses the "largest and most diverse botanical and mycological collections in the world". …
[4] … The collection of fungi at Kew Gardens includes species such as … "Lepiota viridigleba" … | I'm sorry, but I couldn't find any information about a fungus described by Elsie Maud Wakefield as "Lepiota" in a London botanical garden. I did find that Elsie Maud Wakefield was a mycologist at the Royal Botanic Gardens, Kew from 1910 to 1951 and was a specialist in Basidiomycetes and an international authority on Aphyllophorales [1,2].

[1]: … Elsie Maud Wakefield, was an English mycologist and plant pathologist…
[2]: … Elsie Wakefield was a specialist in Basidiomycetes and an international authority on Aphyllophorales. … |

**Figure 5: Case study of SearChain and New Bing in marking references to supporting documents.**

**Table 5: Evaluation of traceability.**

|  | SKC | AMP |
|---|---|---|
| New Bing | 1.143 | 0.45 |
| SearChain | **2.882** | **0.80** |

**Table 6: Efficiency analysis.**

|  | #n ↓ | #m ↓ | #r ↓ | t(s) ↓ | Perf. (Avg) ↑ |
|---|---|---|---|---|---|
| Self-Ask | 401 | 63 | 2.19 | 6.63 | 46.73 |
| Plan-and-Solve w/ IR | 450 | 71 | 1 | 6.05 | 48.89 |
| React → CoT-SC | 938 | 110 | 2.35 | 8.25 | 48.47 |
| Verify-and-Edit | 565 | 307 | 2.40 | 13.90 | 48.88 |
| Tree-of-Thought w/ IR | 622 | 341 | 2.29 | 13.28 | 50.47 |
| DSP | 1759 | 155 | 2.15 | 10.47 | 50.44 |
| SearChain | 390 | 189 | 2.21 | 8.52 | 53.29 |

complex question (global perspective), rather than answering or generating the sub-questions step by step (local perspective). Case study in Figure 4 shows that CoT and Self-Ask stop the reasoning while SearChain continues reasoning by rewriting the sub-question. More reasoning steps in Table 4 also support that SearChain can try more for difficult sub-questions. More case studies are shown in Section A.1.2 of Appendix.

*4.3.4 **SearChain vs New Bing in Tracing**.* We compare the performance of SearChain and New Bing[4] in marking references for generated contents via case study (Figure 5). We further propose two metrics to evaluate the Scope of Knowledge Coverage and Accuracy of Marking Position to show traceability more intuitively:
- **Scope of Knowledge Coverage (SKC) [0, +]**: The number of knowledge items marked with supporting documents in the generated contents. (statistics)
- **Accuracy of Marking Position (AMP) [0, 1]**: The accuracy of the position of the reference marks. That is, whether the references are correctly marked on the sub-fragments for the corresponding knowledge in the generated contents. (human evaluation)

We introduce three humans with master's degrees to participate in our human evaluation and the results are shown in Table 5. SearChain can mark references for each knowledge involved in the reasoning process (i.e., correct nodes of CoQ) in a fine-grained manner. While the references given by New Bing do not cover all of the knowledge and cannot be marked on the correct position. More case studies are shown in Section A.1.1 of Appendix.

*4.3.5 **Efficiency Analysis**.* We analyze the running efficiency between SearChain and baselines on the number of words in the

---

[4]https://www.bing.com/new

input ($n$) and output ($m$) text of LLM, number of rounds of interaction between LLM and IR ($r$) and overall running time ($t$). Table 6 shows our method significantly improves task performance with no significant increase in time consumption. Most baselines also require multiple rounds of interaction between IR and LLM.

## 5 CONCLUSION

In this paper, we pointed out the challenges of introducing IR into LLM from the perspectives of reasoning and knowledge. We then proposed a novel framework named SearChain to enable IR and LLM to interact with each other effectively. SearChain not only stimulates the knowledge-reasoning ability of LLM but also uses IR to provide the knowledge that LLM really needs based on the external knowledge base, which improves both accuracy and credibility. Besides, SearChain can mark references to supporting documents for the knowledge involved in the generated contents, which improves the traceability of the contents. In addition, the interaction between IR and LLM in SearChain transforms the reasoning path from a chain to node-identify Depth-first Search on a tree, which enables LLM to dynamically modify the reasoning direction. Experimental results on complex knowledge-intensive tasks show that SearChain performs better than all baselines.

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

# A APPENDIX

## A.1 Case Study

In this section, we compare the performance of SearChain and New Bing [5] in adding references to supporting documents for generated content via case study. We also use case study to further analyze why CoQ has stronger reasoning ability than Baselines.

*A.1.1 Case Study for SearChain vs New Bing in Tracing.* We compare the performance of SearChain and New Bing in marking references for generated contents via case study (Figure 6 ∼ 7). SearChain can mark references for each knowledge involved in the reasoning process (i.e., each correct node of CoQ) in a more fine-grained manner. While references given by New Bing do not cover all of the knowledge, and in some cases New Bing cannot find the knowledge. SearChain provides a novel perspective that decomposes complex multi-step knowledge-intensive tasks into multiple single-step knowledge reasoning problems, retrieving the supporting documents of knowledge for each step of reasoning, and organizing these reasoning steps with their reference marks as final generated content. This enables the supporting documents to cover every knowledge involved in the generated content, which enhances the traceability of the generated content.

| The movies The Boatniks and The Great Locomotive Chase were both made by which production company? | |
|---|---|
| **SearChain** | **New Bing** |
| The production company for The Boatniks movie is Walt Disney Productions [1]. The production company for The Great Locomotive Chase movie is Walt Disney Productions [2]. So answer is Walt Disney Productions. | Both "The Boatniks" and "The Great Locomotive Chase" were made by Walt Disney Productions [1]. |
| [1]: … The Boatniks is a 1970 American comedy film starring Robert Morse, Stefanie Powers, Don Ameche and Phil Silvers made by Walt Disney Productions, … 
 [2]: … The Great Locomotive Chase is a 1956 Walt Disney Productions … | [1]: … The Boatniks is a 1970 American comedy film directed by Norman Tokar and starring Robert Morse, Stefanie Powers, Don Ameche and Phil Silvers. It was made by Walt Disney Productions, released by Buena Vista Distribution … |

| What is the 2010 population of the city 2.1 miles southwest of Marietta Air Force Station? | |
|---|---|
| **SearChain** | **New Bing** |
| The city 2.1 miles southwest of Marietta Air Force Station is Smyrna [1]. The population of Smyrna in 2010 was 51, 271 [2]. So the final answer is 51,271. | I'm sorry, but I couldn't find any information about a Marietta Air Force Station or a city located 2.1 miles southwest of it. There is a Dobbins Air Reserve Base located southeast of Marietta, Georgia, but I couldn't find any information about the population of a city located 2.1 miles southwest of it [1]. |
| [1]: … Marietta Air Force Station … It is located 2.1 miles northeast of Smyrna, Georgia … 
 [2]: … Smyrna is a city northwest of the neighborhoods of Atlanta. As of the 2010 census, the city had a population of 51,271 … | [1] … Dobbins Air Reserve Base is a United States Air Force reserve air base located in Marietta, Georgia, a suburb about 20 miles (32 km) northwest of Atlanta… |

| Which lead actor/actress in War Chhod Na Yaar has also acted in Bengali and English-language films? | |
|---|---|
| **SearChain** | **New Bing** |
| The lead actors/actresses in War Chhod Na Yaar are Sharman Joshi, Soha Ali Khan, and Javed Jaffrey [1]. Soha Ali Khan acted in Bengali and English-language films according [2]. So the answer is Soha Ali Khan. | I'm sorry, but I couldn't find any information about any lead actor or actress in War Chhod Na Yaar who has also acted in Bengali and English-language films. The lead actors in War Chhod Na Yaar are Sharman Joshi, Soha Ali Khan, and Javed Jaffrey1 [1]. Is there anything else you would like to know about this topic? |
| [1]: … Joshi, Soha Ali Khan, Javed Jaffery, Sanjai Mishra, Dalip Tahil and Mukul Dev who play important characters in the film … 
 [2]: … Soha Ali Khan Khemu (born Soha Ali Khan , 4 October 1978) is an Indian film actress, … she has also acted in Bengali and English-language films. | [1] War Chhod Na Yaar … lead actors, Sharman Joshi, Soha Ali Khan, Javed Jaffery, Sanjai Mishra, Dalip Tahil, and Mukul Dev. |

| How large is the shopping mall where KGOT radio station has its studios ? | |
|---|---|
| **SearChain** | **New Bing** |
| The studios of KGOT radio station are located in the Dimond Center [1], which is a 728,000 ft² (67,000 m²) shopping mall in Anchorage, Alaska [2]. The size of the shopping mall where KGOT radio station has its studios is 728,000 ft² (67,000 m²). So the answer is 728,000 ft² . | I'm sorry, but I couldn't find any information about the size of the shopping mall where KGOT radio station has its studios. KGOT is a commercial FM radio station in Anchorage, Alaska, and it broadcasts from studios on East Dimond Boulevard in the Dimond Center [1]. |
| [1]: …KGOT (101.3 FM) is a commercial Top 40 (CHR) radio station …Dimond Center. 
 [2]: …Dimond Center is a 728,000 ft² (67,000 m²) shopping mall in Anchorage, Alaska ... | [1]: … KGOT (101.3 MHz) is a commercial FM radio station … on East Dimond Boulevard in the Dimond Center. |

⁵https://www.bing.com/new

| Who is the author of the 1993 production Madge Ryan participated in? | |
|---|---|
| **SearChain** | **New Bing** |
| Madge Ryan was known for her role in Medea in the year 1993 [1]. Medea was written by Euripides [2]. So, the final answer is Euripides. | I'm sorry but I couldn't find any information about the author of the 1993 production Madge Ryan participated in. Could you please provide more information or context so that I can help you better? |
| [1] … Madge Ryan Madge Winifred Ryan was an Australian actress, known for her stage roles in London productions of "Entertaining Mr Sloane" (1964), "Philadelphia, Here I Come" (1967), and "Medea" (1993)… 
 [2] … Medea (Seneca) \| Medea (Seneca) Medea is a "fabula crepidata" (Roman tragedy with Greek subject) of about 1027 lines of verse written by Seneca … | |

| Who is the psychologist who wrote about the disciple of the person known as the "Wild Woman of Buttle"? | |
|---|---|
| **SearChain** | **New Bing** |
| Mary MacLane is known as the "Wild Woman of Buttle" [1]. James G. Kiernan wrote about Mary MacLane's disciple Viola Larsen [2]. So, the final answer is James G. Kiernan. | I'm sorry, but I couldn't find any information about the disciple of the person known as the "Wild Woman of Buttle" or the psychologist who wrote about them. Could you provide more details or perhaps rephrase your question? |
| [1] … Mary MacLane was a controversial Canadian-born. MacLane was known as the "Wild Woman of Butte"… 
 [2] James G. Kiernan was an American psychologist, Kiernan went on to write of a variety of topics, e.g. Mary MacLane's disciple Viola Larsen … | |

| The Dogwoman telemovies were created by an actress born in what year? | |
|---|---|
| **SearChain** | **New Bing** |
| Magda Szubanski played the role in Dogwoman telemovies [1] created in 1999 [2], and she was born on April 12, 1961 [3]. So the answer is 1961. | I'm sorry, but I couldn't find any information about the Dogwoman telemovies or the actress who created them. Could you provide more details or perhaps rephrase your question? |
| [1] … Morice had a starring role in the series of three "Dogwoman" telemovies with Magda Szubanski … 
 [2] … Magda Szubanski … In 1999, Szubanski created, wrote, co-produced and starred as Margaret O'Halloran in the "Dogwoman" series of TV films … 
 [3] … Magda Szubanski was born 12 April 1961 … | |

| Jack Alfred Evans played with a football club team that is based in what city in Australia? | |
|---|---|
| **SearChain** | **New Bing** |
| Jack Alfred Evans played with St Kilda football club team [1], which is located in Victoria [2], and specifically in Melbourne [3]. So the final answer is Melbourne. | Jack Alfred Evans was an Australian rules footballer who played with St Kilda in the Victorian Football League (VFL) [1]. St Kilda Football Club is based in Melbourne, Victoria, Australia. |
| [1] … Jack Evans was an Australian rules footballer who played with St Kilda VFL … 
 [2] … St Kilda, Victoria of Australia is located in Alma Road … 
 [3] … St Kilda is an inner suburb of the metropolitan area of Melbourne, Victoria, Australia … | [1] Jack Alfred Evans (born 11 October 1930) was an Australian rules footballer who played with St Kilda in the Victorian Football League (VFL). |

| "The Best of LeAnn Rimes" contained a song used as the theme song for a film. What is the name of the film? | |
|---|---|
| **SearChain** | **New Bing** |
| The album "The Best of LeAnn Rimes" includes the song "Can't Fight the Moonlight" [1], which is the theme song for the film "Coyote Ugly" [2]. So the answer is "Coyote Ugly". | The song "Can't Fight the Moonlight" by LeAnn Rimes was used as the theme song for the film Coyote Ugly [1]. Is there anything else you would like to know? |
| [1] The Best of LeAnn Rimes is a greatest hits album by American country singer LeAnn Rimes, released in 2004, … ,including "How Do I Live", "Can't Fight the Moonlight" and her duet with Ronan Keating, "Last Thing on My Mind". 
 [2] "Can't Fight the Moonlight" is a song written by Diane Warren, recorded … It was the theme song of the film "Coyote Ugly". | [1] … LeAnn Rimes - Can't Fight The Moonlight (Theme from "Coyote Ugly") [1080p]… |

| When did the tour begin for the album Taylor Swift released on October 22, 2012? | |
|---|---|
| **SearChain** | **New Bing** |
| The name of the album Taylor Swift released on October 22, 2012, is "Red" [1]. "The Red Tour" was organized in support of the album "Red" [2], and it started on March 13, 2013 [3]. So, the final answer is March 13, 2013. | Taylor Swift released her fourth studio album, "Red," on October 22, 2012 [1]. However, I'm sorry but I couldn't find the exact date when the tour for this album began. Is there anything else you would like to know? |
| [1] … Red (Taylor Swift album) … released on October 22, 2012 … 
 [2] The Red Tour The Red Tour was the third concert tour by Taylor Swift. Launched in support of Swift's fourth studio album, "Red" … 
 [3] … The Red Tour … began on March 13, 2013 … | [1] … Red (Taylor Swift album) … released on October 22, 2012 … |

**Figure 6: Case study for SearChain vs New Bing in Tracing.**

**The Duke Steps Out stars an actress who was ranked tenth on a list of greatest female Hollywood stars in what year?**

| ✂ SearChain | ⌕ New Bing |
|---|---|
| Joan Crawford starred in The Duke Steps Out [1], which was released on May 14, 1929 [2]. The American Film Institute (AFI) ranked the greatest female Hollywood stars in 1999 [3], and the top five were Katharine Hepburn, Bette Davis, Audrey Hepburn, Ingrid Bergman, and Greta Garbo [4]. Joan Crawford was ranked tenth on the list [5]. So the answer is 1999. | I'm sorry, but I couldn't find any information about the actress from TheDuke Steps Out who was ranked tenth on a list of greatest female Hollywood stars [1]. Could you provide more details or perhaps rephrase your question? |
| [1] .. The Duke Steps Out The Duke Steps Out is directed by James Cruze and starring William Haines and Joan Crawford … | [1]: AFI's 100 Years…100 Stars is the American Film Institute's list ranking the top 25 male and 25 female screen legends of American film history and is the second list of the AFI 100 Years… series. |
| [2] .. The Duke Steps Out is a 1929 silent comedy-drama film… | |
| [3] … AFI's 100 Years…100 Stars is a list of the top 25 male and 25 female Hollywood stars …. The list was unveiled by the American Film Institute on June 15, 1999 … | |
| [4] … "AFI's 100 Years…100 Stars" … Of the 25 actresses listed, Davis was ranked at number two, behind Katharine Hepburn. The United States Postal Service honored Davis with a commemorative postage stamp in 2008, marking the 100th anniversary of her birth … | |
| [5] … Joan Crawford … was also voted the tenth greatest female star of the classic American cinema by the American Film Institute … | |

**Lorenzo Monaco was influenced by which Italian painter and architect?**

| ✂ SearChain | ⌕ New Bing |
|---|---|
| Lorenzo Monaco lived and worked between 1370 and 1425 [1]. The famous Italian painters and architects during his time included Giotto, Masaccio, Brunelleschi, and Donatello [2]. Giotto and Masaccio specifically influenced his work [2], as well as Piero di Giovanni in Siena, Spinello Aretino, and Agnolo Gaddi [2]. So, the answer to the question "Lorenzo Monaco was influenced by which Italian painter and architect?" is Giotto, Masaccio, Piero di Giovanni in Siena, Spinello Aretino, and Agnolo Gaddi. | Lorenzo Monaco (c. 1370 – c. 1425) was an Italian painter of the late Gothic to early Renaissance age. He was born Piero di Giovanni in Siena, Italy and was apprenticed in Florence. He was influenced by Giotto and that artist's followers Spinello Aretino and Agnolo Gaddi1 [1]. |
| [1] …Lorenzo Monaco Lorenzo Monaco (born Piero di Giovanni; 1370 – c. 1425) was an Italian painter of the … | [1]: Lorenzo Monaco Lorenzo Monaco was an Italian painter of the late Gothic-early Renaissance age. He was born Piero di Giovanni in Siena. Little is known about his youth, apart from the fact that he was apprenticed in Florence. He was influenced by Giotto and his followers Spinello Aretino and Agnolo Gaddi. In 1390 he joined the Camaldolese … |
| [2] Lorenzo was an Italian painter of the late Gothic-early Renaissance age. He was born Piero di Giovanni in Siena. Little is known about his youth, apart from the fact that he was apprenticed in Florence. He was influenced by Giotto and his followers Spinello Aretino and Agnolo Gaddi. In 1390 he joined the Camaldolese monastery of Santa Maria degli Angeli. He was thenceforth generally known as "Lorenzo Monaco" (English: "Lawrence the Monk"). In the 1390s he executed three panels of the Biblioteca Laurenziana for his convent … | |

**The song Arizona was recorded by Paul Revere and Mark Lindsay but who wrote the song?**

| ✂ SearChain | ⌕ New Bing |
|---|---|
| Paul Revere is Patriot in the American Revolution [1]. Mark Lindsay is a musician who is the lead singer of Paul Revere & the Raiders [2]. The song Arizona was recorded by Paul Revere and Mark Lindsay, and written by Kenny Young [3]. So the final answer is Kenny Young. | The song "Arizona" was written by Kenny Young and recorded by Mark Lindsay, a solo effort while still lead singer for Paul Revere and the Raiders in 1969 [1]. Is there anything else you would like to know? |
| [1] .. Paul Revere was an American silversmith, engraver, early industrialist, and Patriot in the American Revolution. .. | [1]: … "Arizona" is a song written by Kenny Young and recorded in 1969 by Mark Lindsay, a solo effort while still lead singer for Paul Revere and the Raiders … |
| [2] … Mark Lindsay (born March 9, 1942) is an American musician, best known as the lead singer of Paul Revere & the Raiders … | |
| [3] … "Arizona" is a song written by Kenny Young and recorded in 1969 by Mark Lindsay, a solo effort while still lead singer for Paul Revere and the Raiders … | |

**When was the album with the song Unbelievable by American rapper The Notorious B.I.G released?**

| ✂ SearChain | ⌕ New Bing |
|---|---|
| The Notorious B.I.G., also known as Christopher Wallace, was an American rapper [1]. He released two studio albums during his lifetime: "Ready to Die" in 1994 and "Life After Death" in 1997 [2]. The song "Unbelievable" was included on his debut album "Ready to Die [3]," which was released on September 13, 1994 [4]. So the answer is September 13, 1994 . | The song "Unbelievable" by American rapper The Notorious B.I.G. was released on his debut studio album Ready to Die on September 13, 1994 by Bad Boy Records and Arista Records [1]. Is there anything else you would like to know? |
| [1] … The Notorious B.I.G. Christopher George Latore Wallace (May 21, 1972 – March 9, 1997), known professionally as the Notorious B.I.G., Biggie Smalls, or Biggie, was an American rapper… | [1] … Ready to Die | Ready to Die Ready to Die is the debut studio album by American rapper The Notorious B.I.G., released on September 13, 1994 … |
| [2] … Life After Death Life After Death is the second and final studio album by American rapper The Notorious B.I.G., released on March 25, 1997 … further delving into the mafioso rap subgenre. The album is a sequel to his first album, "Ready to Die" … | |
| [3] … Ready to Die … Songs on the album range from homicide narratives ("Warning") to braggadocios battle raps ("The What," "Unbelievable") … | |
| [4] … Ready to Die | Ready to Die Ready to Die is the debut studio album by American rapper The Notorious B.I.G., released on September 13, 1994 … | |

**Figure 7: Case study for SearChain vs New Bing in Tracing.**

*A.1.2  Case Study for CoQ vs Baselines in Reasoning.* Baselines focus on solving local sub-questions while ignoring the global planning of the reasoning chain, which leads LLM to tend to stop reasoning rather than try more when a sub-question cannot be solved. In our method, LLM acts as a commander that plans a global reasoning chain that can solve the complex question, when a sub-question cannot be solved, even without the help of IR, LLM can try to further decompose or rewrite the sub-question to continue reasoning. It is because our method focuses on building a global chain that can solve the complex question (global perspective), rather than answering or generating sub-questions step by step (local perspective). This makes LLM try more when faced with intermediate difficulties to finally solve complex questions. Case study shown in Figure 8 ~ 9 indicates that SearChain continues reasoning but baselines stop.

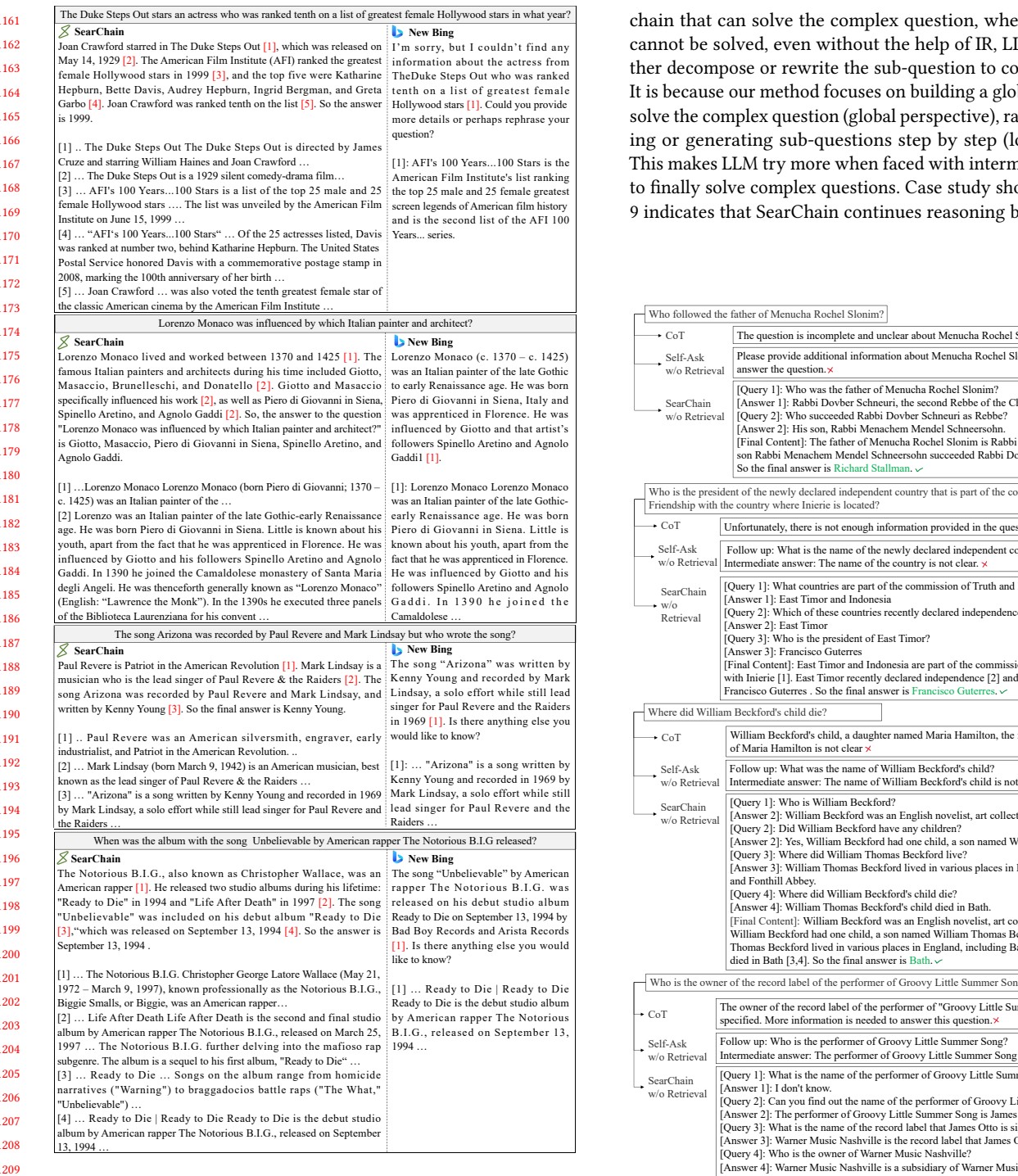

**Figure 8: Case study for CoQ vs Baselines in Reasoning.**

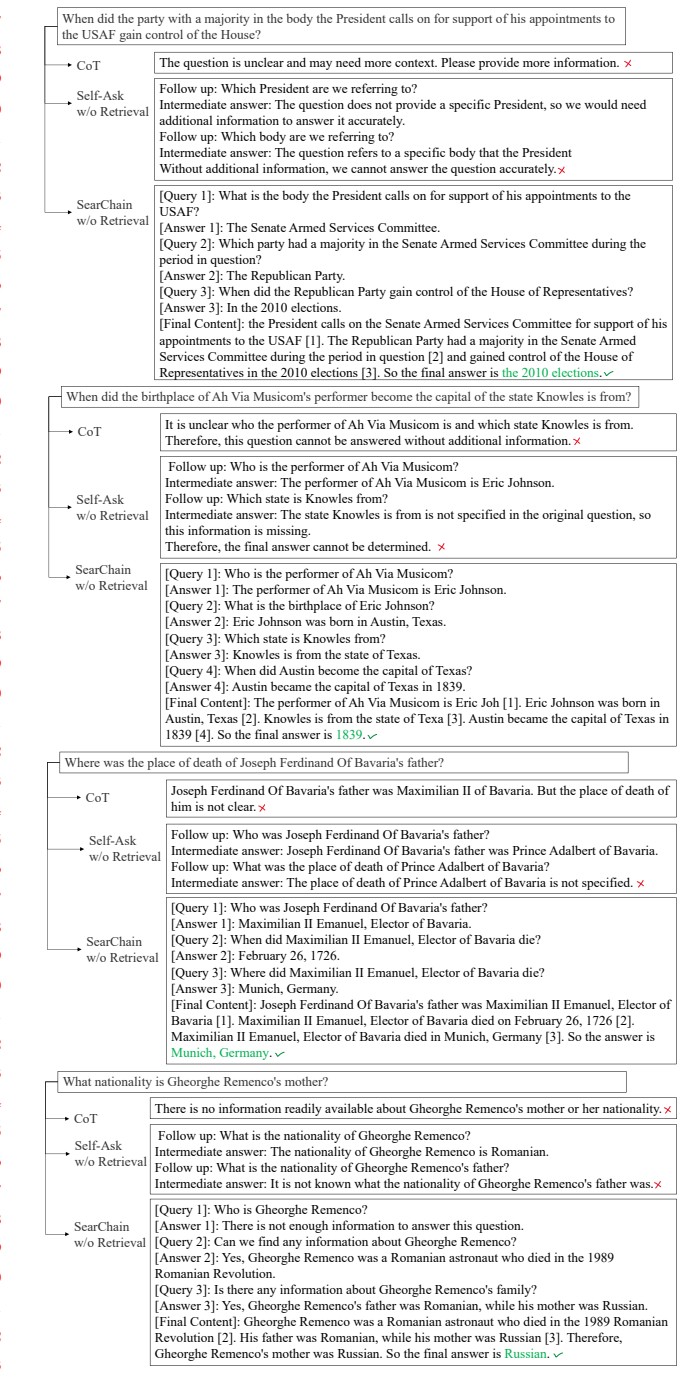

**Figure 9: Case study for CoQ vs Baselines in Reasoning.**

## A.2 Analysis of IR for LLM in Complex Knowledge-Intensive Tasks

In this section, we analyze the effect of IR on LLM on complex knowledge-intensive tasks from perspectives of knowledge-reasoning ability and conflict in the knowledge of IR and LLM.

*A.2.1 Knowledge-reasoning Ability Affected by the Existing Form of Knowledge.* We divide the knowledge of LLM into two forms of existence, one exists in the parameters, and the other exists in the input prompt. In this section, we explore the effect of two different forms on the knowledge-reasoning ability of LLM. The experiments are performed on *gpt-3.5-turbo*.

**Table 7: Accuracy on $\mathbb{S}_{all}$**

|          | 2-Hop | 3-Hop | 4-Hop |
|----------|-------|-------|-------|
| Accuracy | 0.394 | 0.317 | 0.093 |

Specifically, we obtain the sub-questions corresponding to each complex question from the datasets provided by Musique, and then we select the complex questions ($\mathbb{S}_{all}$) that LLM can answer all sub-questions of them from full datasets. Table 7 shows the accuracy of LLM in answering the complex questions in $\mathbb{S}_{all}$, which indicates that even though the parameters have memorized the answer to each sub-question of the complex question, LLM cannot compose these answers to effectively reason the complex questions.

Furthermore, we explore the effect of the form of knowledge on reasoning ability. Specifically, we select the complex questions ($\mathbb{S}_i$) that LLM can answer the sub-questions for the last $i$ steps. And input the answers of sub-questions for the first $n - i$ steps ($n$ is the number of sub-questions) in the form of prompt. In this way, when LLM reasoning the complex questions, the knowledge of $n - i$ sub-questions comes from the prompt, and the knowledge of $i$ sub-questions comes from the parameters. Figure 10(a) shows the performance varies with $i$, which indicates that when $i$ increases, the knowledge in prompt decreases, the knowledge in parameters increases, and the accuracy of LLM decreases. To prevent the impact of the leakage of sub-questions in the prompt, we also conduct the same experiment on $\mathbb{S}_{all}$, except that the prompt only contains sub-questions without the answers (knowledge is still from parameters). The results are shown in Figure 10(b). The accuracy of Figure 10(b) is lower than that in Figure 10(a), which further confirms that the LLM has the stronger knowledge-reasoning ability for complex questions when the knowledge exists in the prompt and shows that the paradigm of using IR to provide knowledge for LLM can not only help LLM acquire its unknown knowledge but also improve the knowledge-reasoning ability of LLM.

*A.2.2 Conflict in the Knowledge of IR and LLM.* Although IR can provide additional knowledge to LLM, contradictory knowledge in IR and LLM can negatively affect the performance of LLM. We use five Open-domain question-answering datasets (Natural Questions, TriviaQA, WebQuestions, SQuAD, and HotpotQA) to detect the knowledge in LLM and IR. We select the set of questions that LLM gives correct answers ($\mathbb{S}_t$) and the set that LLM gives wrong answers ($\mathbb{S}_f$). We evaluate the probability that IR can rank the correct document at Top-1 on $\mathbb{S}_t$ and $\mathbb{S}_f$ respectively. The results shown in Table 8 indicate that although IR can provide LLM with its unknown or wrong knowledge (Top-1 on $\mathbb{S}_f$), it can also interfere with correct knowledge in LLM because Top-1 on $\mathbb{S}_t$ only reach ($60\% \sim 73\%$) and accuracy of LLM drops from $100\%$ to ($80\% \sim 90\%$). This indicates that decoupling the knowledge of LLM and IR in

**Table 8: Top-1 on $\mathbb{S}_t$ and $\mathbb{S}_f$. IR model is ColBERTv2. Red means worse performance after adding IR, green means better performance after adding IR.**

|  | NQ$_{\mathbb{S}_t}$ | NQ$_{\mathbb{S}_f}$ | WQ$_{\mathbb{S}_t}$ | WQ$_{\mathbb{S}_f}$ | Trivia$_{\mathbb{S}_t}$ | Trivia$_{\mathbb{S}_f}$ | Squad$_{\mathbb{S}_t}$ | Squad$_{\mathbb{S}_f}$ | HotpotQA$_{\mathbb{S}_t}$ | HotpotQA$_{\mathbb{S}_f}$ |
|---|---|---|---|---|---|---|---|---|---|---|
| IR Top-1 | 60.99 | 29.20 | 63.24 | 26.07 | 72.60 | 28.10 | 65.88 | 40.67 | 55.61 | 29.13 |
| LLM Acc. | 100 | 0 | 100 | 0 | 100 | 0 | 100 | 0 | 100 | 0 |
| LLM w/ IR Acc. | 80.75 | 25.43 | 85.34 | 23.17 | 89.47 | 25.32 | 86.45 | 30.42 | 80.42 | 19.86 |

**Table 9: Performance of SearChain and DSP on complex knowledge-intensive tasks on Vicuna-13B. Bold denotes the best result in different settings. FC: Fact Checking, LFQA: Long-Form QA. Metric for LFQA: ROUGE-L. Metric for others: cover-EM.**

|  | Muti-Hop QA | | | | Slot Filling | | FC | LFQA |
|---|---|---|---|---|---|---|---|---|
|  | HoPo | MQ | WQA | SQA | zsRE | T-REx | FEV. | ELI5 |
| | Interaction with Information Retrieval | | | | | | | |
| DSP | 25.45 | 9.06 | 27.50 | 62.01 | 33.71 | 49.08 | 73.05 | 22.58 |
| SearChain | **29.77** | **10.59** | **32.32** | **63.75** | **36.86** | **52.75** | **75.47** | **24.05** |

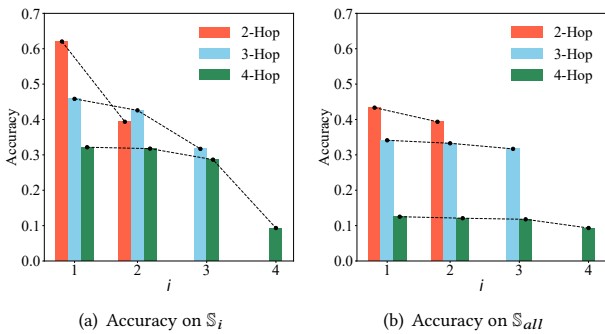

(a) Accuracy on $\mathbb{S}_i$    (b) Accuracy on $\mathbb{S}_{all}$

**Figure 10: Accuracy varies with knowledge form. The larger $i$ is, the larger the proportion of knowledge comes from parameters, and the smaller the proportion of knowledge comes from prompt.**

an explainable way that only provides LLM with its unknown or wrong knowledge to avoid misleading LLM is important.

## A.3 Performance on Vicuna-13B

In this section, we compare SearChain with the competitive baseline DSP on Vicuna-13B [6], a strong open source large model [7] trained by Stanford. The experimental results in Table 9 show that SearChain again outperforms DSP on Vicuna-13B.

## A.4 Experimental Details

*A.4.1 Threshold Selection.* As for the confidence threshold ($\theta$), we initialize the initial value of the confidence threshold (1.0) based on prior knowledge and gradually increase the value with a step size of 0.1. We validate the F1-score (a comprehensive metric of the Recall and Precision of judging whether the passage can answer the question) on the mixed open-domain QA datasets (NQ, TriviaQA, WebQ, and TREC) after each value change. We find that when the

---

[6]https://lmsys.org/blog/2023-03-30-vicuna/
[7]https://huggingface.co/lmsys/vicuna-13b-delta-v1.1/tree/main

**Table 10: Performance change with ROUGE threshold.**

|  | $\alpha = 0.30$ | $\alpha = 0.35$ | $\alpha = 0.40$ | $\alpha = 0.45$ | $\alpha = 0.50$ |
|---|---|---|---|---|---|
| Performance | 25.50 | 25.57 | 25.58 | 25.57 | 25.55 |

confidence threshold is 1.5, the highest F1-score can be achieved so we set the confidence threshold as 1.5. As for the ROUGE threshold ($\alpha$), we determine this value by manually observing the ROUGE relationship between the generated text and the ground truth in the few examples in in-context learning. Our further experiments in Table 10 show that when the value range of ROUGE threshold is between 0.3 and 0.5, the performance change on ELI5 is not obvious.

*A.4.2 Number of Examples in Prompt.* We show the number of examples in prompt used for in-context learning on different datasets (Table 11). Our method (SearChain) achieves the best performance with fewer learning examples than competitive baselines.

```
Construct a global reasoning chain for this complex question [Question]:"{}" and answer the question, and generate a query to the
search engine based on what you already know at each step of the reasoning chain, starting with [Query].
You should generate the answer for each [Query], starting with [Answer].
You should generate the final answer for the [Question] by referring the [Query]-[Answer] pairs, starting with [Final Content].

For exmaple:
[Question]:"How many places of higher learning are in the city where the Yongle emperor greeted the person to whom the edict
was addressed?"
[Query 1]: Who was the edict addressed to?
[Answer 1]: the Karmapa
[Query 2]: Where did the Yongle Emperor greet the Karmapa?
[Answer 2]: Nanjing
[Query 3]: How many places of higher learning are in Nanjing?
[Answer 3]: 75
[Final Content]: The edict was addressed to Karmapa [1]. Yongle Emperor greet the Karampa in Nanjing [2]. There are 75 places
of higher learning are in Nanjing [3]. So the final answer is 75.

[Question]:"Which magazine was started first Arthur's Magazine or First for Women?"
[Query 1]: When was Arthur's Magazine started?
[Answer 1]: 1844.
[Query 2]: When was First for Women started?
[Answer 2]: 1989
[Final Content]: Arthur's Magazine started in 1844 [1]. First for Women started in 1989 [2]. So Arthur's Magazine was started
first. So the final answer is Arthur's Magazine.
```

**Figure 11: Prompt for generating Chain-of-Query on HotpotQA, Musique, WikiMultiHopQA, zsRE and T-REx (in the setting without information retrieval).**

*A.4.3 Prompts in Experiment.* We show the prompt used in experiment on different datasets in Figure 11 ∼ 13.

**Table 11: Number of examples in prompt used for in-content learning on different datasets.**

| | Muti-Hop QA | | | | Slot Filling | | FC | LFQA |
|---|---|---|---|---|---|---|---|---|
| | HoPo | MQ | WQA | SQA | zsRE | T-REx | FEV. | ELI5 |
| Without Information Retrieval | | | | | | | | |
| Direct Prompting | 0 | 0 | 0 | 0 | 0 | 0 | 0 | 0 |
| Auto-CoT | 4 | 4 | 4 | 6 | 4 | 4 | 4 | 2 |
| CoT | 4 | 4 | 4 | 6 | 4 | 4 | 4 | 2 |
| CoT-SC | 4 | 4 | 4 | 6 | 4 | 4 | 4 | 2 |
| Recite-and-answer | 4 | 4 | 4 | 6 | 4 | 4 | 4 | 2 |
| Self-Ask w/o IR | 4 | 4 | 4 | 6 | 4 | 4 | 4 | 2 |
| Least-to-Most | 4 | 4 | 4 | 6 | 4 | 4 | 4 | 2 |
| Plan-and-Solve | 4 | 4 | 4 | 6 | 4 | 4 | 4 | 2 |
| SearChain w/o IR | 2 | 2 | 2 | 6 | 2 | 2 | 4 | 2 |
| Interaction with Information Retrieval | | | | | | | | |
| Direct Retrieval | 0 | 0 | 0 | 0 | 0 | 0 | 0 | 0 |
| ToolFormer | 4 | 4 | 4 | 6 | 4 | 4 | 4 | 2 |
| Self-Ask | 4 | 4 | 4 | 6 | 4 | 4 | 4 | 2 |
| Plan-and-Solve w/ IR | 4 | 4 | 4 | 6 | 4 | 4 | 4 | 2 |
| React → CoT-SC | 6 | 4 | 4 | 6 | 4 | 4 | 4 | 2 |
| Verify-and-Edit | 2 | 2 | 2 | 2 | 2 | 2 | 4 | 2 |
| Tree-of-Thought w/ IR | 4 | 4 | 4 | 6 | 4 | 4 | 4 | 2 |
| DSP | 16 | 8 | 8 | 8 | 8 | 8 | 8 | 2 |
| SearChain | 2 | 2 | 2 | 2 | 2 | 2 | 4 | 2 |

---

Construct a global reasoning chain for this complex question [Question]:"{}" and answer the question, and generate a query to the search engine based on what you already know at each step of the reasoning chain, starting with [Query].
You should generate the answer for each [Query], starting with [Answer].
You should generate the final answer for the [Question] by referring the [Query]-[Answer] pairs, starting with [Final Content].
For exemple:
[Question]:"What causes the trail behind jets at high altitude?"
[Query 1]: What is the trail behind jets at high altitude?
[Answer 1]: The trail behind jets at high altitude is commonly referred to as a contrail, which is short for "condensation trail." It is formed when the hot exhaust gases from a jet engine mix with the colder air at high altitudes, causing the water vapor in the air to condense and freeze into tiny ice crystals.
[Query 2]: Why do the hot exhaust gases mix with the colder air at high altitudes?
[Answer 2]: The hot exhaust gases from a jet engine mix with the colder air at high altitudes due to the pressure difference between the engine exhaust and the surrounding air. At high altitudes, the air is much colder and thinner, which causes the exhaust gases to rapidly expand and cool, creating a turbulent wake behind the aircraft.
[Query 3]: Why does the water vapor in the air condense and freeze into ice crystals?
[Answer 3]: The water vapor in the air condenses and freezes into ice crystals because the exhaust gases from the jet engine contain a significant amount of water vapor, which is heated to high temperatures in the engine. When the hot water vapor mixes with the cold air at high altitudes, it quickly loses heat and energy, causing the water vapor to condense into liquid droplets and then freeze into ice crystals.
[Final Content]: The trail behind jets at high altitude is formed when the hot exhaust gases from a jet engine mix with the colder air at high altitudes, causing the water vapor in the air to condense and freeze into tiny ice crystals [1]. The exhaust gases mix with the colder air due to the pressure difference between the engine exhaust and the surrounding air [2], and the water vapor condenses and freezes because it is heated to high temperatures in the engine and then quickly loses heat and energy when mixed with the cold air at high altitudes [3].

[Question]: In Trading Places (1983, Akroyd/Murphy) how does the scheme at the end of the movie work? Why would buying a lot of OJ at a high price ruin the Duke Brothers?
[Query 1]: What is the scheme at the end of Trading Places?
[Answer 1]: In the movie, the main character, Billy Ray Valentine (Eddie Murphy), and his partner, Louis Winthorpe III (Dan Aykroyd), execute a plan to bankrupt the Duke Brothers by manipulating the frozen concentrated orange juice (FCOJ) futures market.
[Query 2]: How do Billy Ray and Louis manipulate the FCOJ futures market?
[Answer 2]: Billy Ray and Louis obtain insider information about the Department of Agriculture's upcoming crop report, which indicates that a harsh winter has destroyed much of the orange crop. They use this information to purchase FCOJ futures contracts at a low price before the report is released, and then they sell the contracts at a high price after the report's release, when the market has responded to the news of the crop damage.
[Query 3]: Why does buying a lot of OJ at a high price ruin the Duke Brothers?
[Answer 3]: The Duke Brothers, who are also investing in the FCOJ market, have bet that the orange crop will be abundant and that the price of FCOJ will remain low. However, Billy Ray and Louis's scheme drives up the price of FCOJ, causing the Duke Brothers to lose a significant amount of money and ultimately leading to their downfall.
[Final Content]: In Trading Places, Billy Ray Valentine and Louis Winthorpe III manipulate the FCOJ futures market [1] by obtaining insider information about the crop report and purchasing contracts at a low price before selling them at a higher price after the report is released [2]. The Duke Brothers, who have also invested in the market, lose money because they bet on an abundant orange crop and low FCOJ prices. However, Billy Ray and Louis's scheme causes the price of FCOJ to rise, which ruins the Duke Brothers and leads to their downfall [3].

**Figure 12: Prompt for generating Chain-of-Query on ELI5 (in the setting without information retrieval).**

---

Construct a global reasoning chain for this complex question [Question]:"{}" and answer the question, and generate a query to the search engine based on what you already know at each step of the reasoning chain, starting with [Query].
You should generate the answer for each [Query], starting with [Answer].
You should generate the final answer for the [Question] by referring the [Query]-[Answer] pairs, starting with [Final Content].
If you don't know the answer, generate a query to the search engine based on what you already know and donot know, starting with [Unsolved Query] and please stop your generation.
For exemple:
[Question]:"How many places of higher learning are in the city where the Yongle emperor greeted the person to whom the edict was addressed?"
[Query 1]: Who was the edict addressed to?
If you don't know the answer:
[Unsolved Query]: Who was the edict addressed to?
If you know the answer:
[Answer 1]: the Karmapa
[Query 2]: Where did the Yongle Emperor greet the the Karmapa?
If you don't know the answer:
[Unsolved Query]: here did the Yongle Emperor greet the the Karmapa?
If you know the answer:
[Answer 2]: Nanjing
[Query 3]: How many places of higher learning are in Nanjing ?
If you don't know the answer:
[Unsolved Query]: How many places of higher learning are in Nanjing?
If you know the answer:
[Answer 3]: 75
[Final Content]: The edict was addressed to Karmapa [1]. Yongle Emperor greet the Karmapa in Nanjing [2]. There are 75 places of higher learning are in Nanjing [3]. So the final answer is 75.

[Question]:"Nicholas Brody is a character on Homeland. (SUPPORTS or REFUTES)?"
[Query 1]: What is Homeland?
[Answer 1]: Homeland is a television series.
[Query 2]: Is Nicholas Brody a character in Homeland?
[Answer 2]: Yes.
[Final Content]: Homeland is a television series [1]. Nicholas Brody is a character in Homeland [2]. So the final answer is SUPPORTS.

[Question]:"Brad Wilk helped co-found Rage in 1962. (SUPPORTS or REFUTES)?"
[Query 1]: Did Brad Wilk co-found Rage?
[Answer 1]: Yes
[Query 2]: Did Brad Wilk co-found Rage in 1962?
[Answer 2]: No, Rage was founded in 1991
[Final Content]: Brad Wilk did co-found Rage [1], but not in 1962 [2]. So the final answer is REFUTES.

[Question]:"Aristotle spent time in Athens. (SUPPORTS or REFUTES)?"
[Query 1]: Who is Aristotle?
[Answer 1]: Aristotle was a Greek philosopher.
[Query 2]: Did Aristotle spend time in Athens?
[Answer 2]: Yes, Aristotle studied and taught at the Academy in Athens for 20 years.
[Final Content]: Aristotle was a Greek philosopher who studied and taught at the Academy in Athens for 20 years [2]. So the final answer is SUPPORTS.

[Question]:"Telemundo is a English-language television network. (SUPPORTS or REFUTES)?"
[Query 1]: What is Telemundo?
[Answer 1]: Telemundo is a television network.
[Query 2]: Is Telemundo an English-language television network?
[Answer 2]: No, Telemundo is a Spanish-language television network.
[Final Content]: Telemundo is a television network [1], but it is not an English-language television network [2]. So the final answer is REFUTES.

**Figure 13: Prompt for generating Chain-of-Query at the first round on FEVER (in the setting with information retrieval).**

Construct a global reasoning chain for this complex question [Question]:"{}" and answer the question, and generate a query to the search engine based on what you already know at each step of the reasoning chain, starting with [Query].
You should generate the answer for each [Query], starting with [Answer].
You should generate the final answer to judge whether to SUPPORTS or REFUTES for the [Question] by referring the [Query]-[Answer] pairs, starting with [Final Content].
For exmaple:

[Question]:"SUPPORTS or REFUTES this claim?: Nicholas Brody is a character on Homeland. (SUPPORTS or REFUTES)?"
[Query 1]: What is Homeland?
[Answer 1]: Homeland is a television series.
[Query 2]: Is Nicholas Brody a character in Homeland?
[Answer 2]: Yes.
[Final Content]: Homeland is a television series [1]. Nicholas Brody is a character in Homeland [2]. So the final answer is SUPPORTS.

[Question]:"SUPPORTS or REFUTES this claim?: Brad Wilk helped co-found Rage in 1962. (SUPPORTS or REFUTES)?"
[Query 1]: Did Brad Wilk co-found Rage?
[Answer 1]: Yes
[Query 2]: Did Brad Wilk co-found Rage in 1962?
[Answer 2]: No, Rage was founded in 1991
[Final Content]: Brad Wilk did co-found Rage [1], but not in 1962 [2]. So the final answer is REFUTES.

[Question]:"SUPPORTS or REFUTES this claim?: Aristotle spent time in Athens. (SUPPORTS or REFUTES)?"
[Query 1]: Who is Aristotle?
[Answer 1]: Aristotle was a Greek philosopher.
[Query 2]: Did Aristotle spend time in Athens?
[Answer 2]: Yes, Aristotle studied and taught at the Academy in Athens for 20 years.
 [Final Content]: Aristotle was a Greek philosopher who studied and taught at the Academy in Athens for 20 years [2]. So the final answer is SUPPORTS.

[Question]:"SUPPORTS or REFUTES this claim?: Telemundo is a English-language television network. (SUPPORTS or REFUTES)?"
[Query 1]: What is Telemundo?
[Answer 1]: Telemundo is a television network.
[Query 2]: Is Telemundo an English-language television network?
[Answer 2]: No, Telemundo is a Spanish-language television network.
[Final Content]: Telemundo is a television network [1], but it is not an English-language television network [2]. So the final answer is REFUTES.

**Figure 14: Prompt for generating Chain-of-Query on FEVER (in the setting without information retrieval).**

Construct a global reasoning chain for this complex [Question] : " {} " You should generate a query to the search engine based on what you already know at each step of the reasoning chain, starting with [Query].
If you know the answer for [Query], generate it starting with [Answer].
You can try to generate the final answer for the [Question] by referring to the [Query]-[Answer] pairs, starting with [Final Content].
If you don't know the answer, generate a query to search engine based on what you already know and do not know, starting with [Unsolved Query].

For example:
[Question]:"Is it common to see frost during some college commencements?"
[Query 1]: What seasons can you expect see frost?
If you don't know the answer:
[Unsolved Query]: What seasons can you expect see frost?
Instruction: Please Stop your generation.
If you know the answer:
[Answer 1]: Winter.
[Query 2]: What months do college commencements occur?
If you don't know the answer:
[Unsolved Query]: What months do college commencements occur?
Instruction: Please Stop your generation.
If you know the answer:
[Answer 2]: December, May, and sometimes June.
[Query 3]: Do any of December, May, and sometimes June occur during winter?
If you don't know the answer:
[Unsolved Query]: Do any of December, May, and sometimes June occur during winter?
If you know the answer:
[Answer 3]: December
Instruction: Please Stop your generation.
If you know the answer:
[Question]:"Is it common to see frost during some college commencements?" (The answer can only be "Yes" or "No")
[Final Content]: You expect see frost in Winter [1]. College commencements occur on December, May, and sometimes June [2]. December, May, and sometimes June occur during winter [3]. So the final answer is Yes.

[Question]:"Would a pear sink in water?"
[Query 1]: What is the density of a pear?
[Answer 1]: 0.59 g/cm^3
[Query 2]: What is the density of water?
[Answer 2]: 1 g/cm^3
[Query 3]: Is 0.59 g/cm^3 greater than  g/cm^3?
[Answer 3]: No
[Question]:"Would a pear sink in water?" (Yes or No)
[Final Content]: The density of a pear is 0.59 g/cm^3 [1]. The density of water is 1 g/cm^3 [2]. 0.59 g/cm^3 is not greater than g/cm^3 [3]. So the final answer is No.

**Figure 15: Prompt for generating Chain-of-Query at the first round on StragegyQA (in the setting with information retrieval).**

Construct a global reasoning chain for this complex question [Question] : " {} " You should generate a query to the search engine based on what you already know at each step of the reasoning chain, starting with [Query].
If you know the answer for [Query], generate it starting with [Answer].
You can try to generate the final answer for the [Question] by referring to the [Query]-[Answer] pairs, starting with [Final Content].
If you don't know the answer, generate a query to search engine based on what you already know and do not know, starting with [Unsolved Query].

For example:
[Question]: "Where do greyhound buses that are in the birthplace of Spirit If...'s performer leave from? "
[Query 1]: Who is the performer of Spirit If... ?
If you don't know the answer:
[Unsolved Query]: Who is the performer of Spirit If... ?
If you know the answer:
[Answer 1]: The performer of Spirit If… is Kevin Drew.
[Query 2]: Where was Kevin Drew born?
If you don't know the answer:
[Unsolved Query]: Where was Kevin Drew born?
If you know the answer:
[Answer 2]: Toronto.
[Query 3]: Where do greyhound buses in Toronto leave from?
If you don't know the answer:
[Unsolved Query]: Where do greyhound buses in Toronto leave from?
If you know the answer:
[Answer 3]: Toronto Coach Terminal.
[Final Content]: The performer of Spirit If… is Kevin Drew [1]. Kevin Drew was born in Toronto [2]. Greyhound buses in Toronto leave from Toronto
Coach Terminal [3]. So the final answer is Toronto Coach Terminal.

[Question]:"Which magazine was started first Arthur's Magazine or First for Women?"
[Query 1]: When was Arthur's Magazine started?
[Answer 1]: 1844.
[Query 2]: When was First for Women started?
[Answer 2]: 1989
[Final Content]: Arthur's Magazine started in 1844 [1]. First for Women started in 1989 [2]. So Arthur's Magazine was started first. So the answer is Arthur's Magazine.

**Figure 16: Prompt for generating Chain-of-Query at the first round on HotpotQA, Musique, WikiMultiHopQA, zsRE and T-REx (in the setting with information retrieval).**

Construct a global reasoning chain for this complex question [Question]:"{}" and answer the question, and generate a query to the search engine based on what you already know at each step of the reasoning chain, starting with [Query].
You should generate the answer for each [Query], starting with [Answer].
You should generate the final answer to judge whether to SUPPORTS or REFUTES for the [Question] by referring the [Query]-[Answer] pairs, starting with [Final Content].
For exmaple:

[Question]:"SUPPORTS or REFUTES this claim?: Nicholas Brody is a character on Homeland. (SUPPORTS or REFUTES)?"
[Query 1]: What is Homeland?
[Answer 1]: Homeland is a television series.
[Query 2]: Is Nicholas Brody a character in Homeland?
[Answer 2]: Yes.
[Final Content]: Homeland is a television series [1]. Nicholas Brody is a character in Homeland [2]. So the final answer is SUPPORTS.

[Question]:"SUPPORTS or REFUTES this claim?: Brad Wilk helped co-found Rage in 1962. (SUPPORTS or REFUTES)?"
[Query 1]: Did Brad Wilk co-found Rage?
[Answer 1]: Yes
[Query 2]: Did Brad Wilk co-found Rage in 1962?
[Answer 2]: No, Rage was founded in 1991
[Final Content]: Brad Wilk did co-found Rage [1], but not in 1962 [2]. So the final answer is REFUTES.

[Question]:"SUPPORTS or REFUTES this claim?: Aristotle spent time in Athens. (SUPPORTS or REFUTES)?"
[Query 1]: Who is Aristotle?
[Answer 1]: Aristotle was a Greek philosopher.
[Query 2]: Did Aristotle spend time in Athens?
[Answer 2]: Yes, Aristotle studied and taught at the Academy in Athens for 20 years.
 [Final Content]: Aristotle was a Greek philosopher who studied and taught at the Academy in Athens for 20 years [2]. So the final answer is SUPPORTS.

[Question]:"SUPPORTS or REFUTES this claim?: Telemundo is a English-language television network. (SUPPORTS or REFUTES)?"
[Query 1]: What is Telemundo?
[Answer 1]: Telemundo is a television network.
[Query 2]: Is Telemundo an English-language television network?
[Answer 2]: No, Telemundo is a Spanish-language television network.
[Final Content]: Telemundo is a television network [1], but it is not an English-language television network [2]. So the final answer is REFUTES.

**Figure 17: Prompt for generating Chain-of-Query on FEVER (in the setting without information retrieval).**

Construct a global reasoning chain for this complex question [Question]:"{}" and answer the question, and generate a query to the search engine based on what you already know at each step of the reasoning chain, starting with [Query].
You should generate the answer for each [Query], starting with [Answer].
You should generate the final answer for the [Question] by referring the [Query]-[Answer] pairs, starting with [Final Content].

For example1:
[Question]:"Do hamsters provide food for any animals?"
[Query 1]: What types of animal are hamsters?
[Answer 1]: Hamsters are prey animals.
[Query 2]: Do prey provide food for any other animals?
 [Answer 2]: Yes
 [Question]:"Do hamsters provide food for any animals?" (Yes or No)
 [Final Content]: Hamsters are prey animals [1]. Prey provide food for other animals [2]. So the final answer is Yes.

For example2:
[Question]:"Could Brooke Shields succeed at University of Pennsylvania?"
[Query 1]: What college did Brooke Shields go to?
[Answer 1]: Princeton University
[Query 2]: Out of all colleges in the US, how is Princeton University ranked"
[Answer 2]: Princeton is ranked as the number 1 national college by US news.
[Query 3]: Is the ranking of University of Pennsylvania similar to Princeton University?
[Answer 3]: Yes
[Question]:"Could Brooke Shields succeed at University of Pennsylvania?" (Yes or No)
[Final Content]: Brooke Shields went to Princeton University [1].  Princeton is ranked as the number 1 national college by US news [2]. The ranking of University of Pennsylvania similar to Princeton University [3]. So the final answer is Yes.

For example3:
[Question]:"Is it common to see frost during some college commencements?"
[Query 1]: What seasons can you expect see frost?
[Answer 1]: Winter.
[Query 2]: What months do college commencements occur?
[Answer 2]: December, May, and sometimes June.
[Query 3]: Do any of December, May, and sometimes June occur during winter?
[Answer 3]: Yes.
[Question]:"Is it common to see frost during some college commencements?" (Yes or No)
[Final Content]: You expect see frost in Winter [1]. College commencements occur on December, May, and sometimes June [2]. December, May, and sometimes June occur during winter [3]. So the final answer is Yes.

For example4:
[Question]:"Would a pear sink in water?"
[Query 1]: What is the density of a pear?
[Answer 1]: 0.59 g/cm^3
[Query 2]: What is the density of water?
[Answer 2]: 1 g/cm^3
[Query 3]: Is 0.59 g/cm^3 greater than  g/cm^3?
[Answer 3]: No
[Question]:"Would a pear sink in water?" (Yes or No)
[Final Content]: The density of a pear is 0.59 g/cm^3 [1]. The density of water is 1 g/cm^3 [2]. 0.59 g/cm^3 is not greater than 1 g/cm^3 [3]. So the final answer is No.

For example5:
[Question]:"Could a llama birth twice during War in Vietnam (1945-46)?"
[Query 1]: How long did the Vietnam war last?
[Answer 1]: Around 6 months.
[Query 2]: How long is llama gestational period?
[Answer 2]: The gestation period for a llama is 11 months.
[Query 3]: What is 2 times 11 months?
[Answer 3]: 22 months.
[Query 4]: Is 6 months longer than 22 months?
[Answer 4]: No
[Question]:"Could a llama birth twice during War in Vietnam (1945-46)?" (Yes or No)
[Final Content]: Vietnam war last around 6 months [1]. The gestation period for a llama is 11 months [2]. 2 times 11 months is 22 months [3]. 6 months is not longer than 22 months [4]. So the final answer is No.

For example6:
[Question]:"Hydrogen's atomic number squared exceeds number of Spice Girls?"
[Query 1]: What is the atomic number of hydrogen?
[Answer 1]: Hydrogen is the first element and has an atomic number of one.
[Query 2]: How many people are in the Spice Girls band?
[Answer 2]: The Spice Girls has five members.
[Query 3]: Is the square of 1 greater than 5?
[Answer 3]: No
[Question]:"Hydrogen's atomic number squared exceeds number of Spice Girls?" (Yes or No)
[Final Content]: Hydrogen is the first element and has an atomic number of one [1]. The Spice Girls has five members [2]. The square of 1 is not greater than 5 [3]. So the final answer is No.

**Figure 18: Prompt for generating Chain-of-Query on Strate-gyQA (in the setting without information retrieval).**

## A.5 Example of the Interaction between IR and LLM

We use Figure 20 and Figure 21 as an example to show the details of the interaction between IR and LLM. In this example, IR and LLM perform three rounds of interaction. IR verifies and corrects the answers to the first two sub-questions and provides the answer to the last sub-question. In each round, LLM re-generate the new CoQ according to the feedback of IR.

Construct a global reasoning chain for this complex [Question] : " {} " You should generate a query to the search engine based on what you already know at each step of the reasoning chain, starting with [Query].
If you know the answer for [Query], generate it starting with [Answer].
You can try to generate the final answer for the [Question] by referring to the [Query]-[Answer] pairs, starting with [Final Content].
If you don't know the answer, generate a query to search engine based on what you already know and do not know, starting with [Unsolved Query].

 For exmaple:
[Question]:"What causes the trail behind jets at high altitude?"
[Query 1]: What is the trail behind jets at high altitude?
If you don't know the answer:
[Unsolved Query]: What is the trail behind jets at high altitude?
If you know the answer:
[Answer 1]: The trail behind jets at high altitude is commonly referred to as a contrail, which is short for "condensation trail." It is formed when the hot exhaust gases from a jet engine mix with the colder air at high altitudes, causing the water vapor in the air to condense and freeze into tiny ice crystals.
[Query 2]: Why do the hot exhaust gases mix with the colder air at high altitudes?
If you don't know the answer:
[Unsolved Query]: Why do the hot exhaust gases mix with the colder air at high altitudes?
If you know the answer:
[Answer 2]: The hot exhaust gases from a jet engine mix with the colder air at high altitudes due to the pressure difference between the engine exhaust and the surrounding air. At high altitudes, the air is much colder and thinner, which causes the exhaust gases to rapidly expand and cool, creating a turbulent wake behind the aircraft.
[Query 3]: Why does the water vapor in the air condense and freeze into ice crystals?
If you don't know the answer:
[Unsolved Query]: Why does the water vapor in the air condense and freeze into ice crystals?
If you know the answer:
[Answer 3]: The water vapor in the air condenses and freezes into ice crystals because the exhaust gases from the jet engine contain a significant amount of water vapor, which is heated to high temperatures in the engine. When the hot water vapor mixes with the cold air at high altitudes, it quickly loses heat and energy, causing the water vapor to condense into liquid droplets and then freeze into ice crystals.
[Final Content]: The trail behind jets at high altitude is formed when the hot exhaust gases from a jet engine mix with the colder air at high altitudes, causing the water vapor in the air to condense and freeze into tiny ice crystals. The exhaust gases mix with the colder air due to the pressure difference between the engine exhaust and the surrounding air, and the water vapor condenses and freezes because it is heated to high temperatures in the engine and then quickly loses heat and energy when mixed with the cold air at high altitudes.

[Question]: In Trading Places (1983, Akroyd/Murphy) how does the scheme at the end of the movie work? Why would buying a lot of OJ at a high price ruin the Duke Brothers?
[Query 1]: What is the scheme at the end of Trading Places?
[Answer 1]: In the movie, the main character, Billy Ray Valentine (Eddie Murphy), and his partner, Louis Winthorpe III (Dan Aykroyd), execute a plan to bankrupt the Duke Brothers by manipulating the frozen concentrated orange juice (FCOJ) futures market.
[Query 2]: How do Billy Ray and Louis manipulate the FCOJ futures market?
[Answer 2]: Billy Ray and Louis obtain insider information about the Department of Agriculture's upcoming crop report, which indicates that a harsh winter has destroyed much of the orange crop. They use this information to purchase FCOJ futures contracts at a low price before the report is released, and then they sell the contracts at a high price after the report's release, when the market has responded to the news of the crop damage.
[Query 3]: Why does buying a lot of OJ at a high price ruin the Duke Brothers?
[Answer 3]: The Duke Brothers, who are also investing in the FCOJ market, have bet that the orange crop will be abundant and that the price of FCOJ will remain low. However, Billy Ray and Louis's scheme drives up the price of FCOJ, causing the Duke Brothers to lose a significant amount of money and ultimately leading to their downfall.
[Final Content]: In Trading Places, Billy Ray Valentine and Louis Winthorpe III manipulate the FCOJ futures market by obtaining insider information about the crop report and purchasing contracts at a low price before selling them at a higher price after the report is released. The Duke Brothers, who have also invested in the market, lose money because they bet on an abundant orange crop and low FCOJ prices. However, Billy Ray and Louis's scheme causes the price of FCOJ to rise, which ruins the Duke Brothers and leads to their downfall.

**Figure 19: Prompt for generating Chain-of-Query at the first round on ELI5 (in the setting with information retrieval).**

Question: Where do greyhound buses that are in the birthplace of Spirit If...'s performer leave from?

**Round 1 LLM CoQ 1 generation**
[Query 1]: "Who is the performer of Spirit If...?"
[Answer 1]: "Casey Morton Cott"
[Query 2]:"What is the place of birth of Casey Morton Cott?"
[Answer 2]: "Cuyahoga County"
[Unsolved Query]:"Where do greyhound buses leave from in Cuyahoga County?"

**Round 1 IR Verification**
For Node 1 of CoQ 1:
[Query 1]: "Who is the performer of Spirit If...?"
[Answer 1]: "Casey Morton Cott"
If Q=Who is the performer of Spirit If...? has been processed ?
→ No
Search ("Q=Who is the performer of Spirit If...?")
Top-1 Passage

Spirit If... is the debut solo album by Kevin Drew. It was released on September 18, 2007 ...
→ Reader Model → Answer is : Kevin Drew
→ Confidence: f
If ("Kevin Drew" not match "Casey Morton Cott" && f > threshold) ?
Yes, construct a prompt to feed back to LLM

According to this Reference, the answer for "Who is the performer of Spirit If...?" should be "Kevin Drew", you can change your answer based on the Reference and continue constructing the reasoning chain to give the final answer for [Question]："Where do greyhound buses that are in the birthplace of Spirit If...'s performer leave from?"
Reference: Spirit If... is the debut solo album by Kevin Drew. It was released on September 18, 2007 ...

**Round 2 LLM CoQ 2 generation**
[Query 1]: "Who is the performer of Spirit If...?"
[Answer 1]: "Kevin Drew"
[Query 2]:"What is the place of birth of Kevin Drew?"
[Answer 2]: "Vancouver"
[Unsolved Query]:"Where do greyhound buses leave from in Vancouver?"

**Round 2 IR Verification**
For Node 1 of CoQ 2:
[Query 1]: "Who is the performer of Spirit If...?"
[Answer 1]: "Kevin Drew"
If Q=Who is the performer of Spirit If...? has been processed ? → Yes
For Node 2 of CoQ 2:
[Query 2]:"What is the place of birth of Kevin Drew?"
[Answer 2]: "Vancouver"
If Q=What is the place of birth of Kevin Drew? has been processed?
→ No
Search ("Q=What is the place of birth of Kevin Drew?"")
Top-1 Passage

... Kevin Drew (born September 9, 1976 in Toronto) ...
→ Reader Model → Answer is : Toronto
→ Confidence: f
If ("Toronto" not match "Vancouver" && f > threshold) ?
Yes, construct a prompt to feed back to LLM

According to this Reference, the answer for "What is the place of birth of Kevin Drew?" should be "Toronto", you can change your answer based on the Reference and continue constructing the reasoning chain to give the final answer for [Question]："Where do greyhound buses that are in the birthplace of Spirit If...'s performer leave from?"
Reference: ... Kevin Drew (born September 9, 1976 in Toronto) ...

**Figure 20: Details (Round 1, 2).**

**Round 3 LLM CoQ 3 generation**
[Query 2]:"What is the place of birth of Kevin Drew?"
[Answer 2]: "Toronto"
[Unsolved Query]:"Where do greyhound buses leave from in Toronto?"

**Round 3 IR Completion**
For Node 1 of CoQ 3:
[Query 2]:"What is the place of birth of Kevin Drew?"
[Answer 2]: "Toronto"
If Q="What is the place of birth of Kevin Drew?" has been processed? → Yes
For Node 2 of CoQ 3:
[Unsolved Query]:"Where do greyhound buses leave from in Toronto?"
If Q="Where do greyhound buses leave from in Toronto?" has been processed?
→ No
Search ("Q=Where do greyhound buses leave from in Toronto?")
Top-1 Passage

The Toronto Coach Terminal is the central bus station for inter-city services in Toronto, Ontario, Canada ... when it was leased out in its entirety to bus lines Coach Canada and Greyhound Canada ...
→ Reader Model → Answer is : Toronto Coach Terminal
→ Confidence: f
construct a prompt to feed back to LLM

According to this Reference, the answer for "Where do greyhound buses leave from in Toronto?" should be "Toronto Coach Terminal ", you can give your answer based on the Reference and continue constructing the reasoning chain to give the final answer for [Question]："Where do greyhound buses that are in the birthplace of Spirit If...'s performer leave from?"
Reference: The Toronto Coach Terminal is the central bus station for inter-city services in Toronto, Ontario, Canada ... when it was leased out in its entirety to bus lines Coach Canada and Greyhound Canada ...

**Final Result: Tracing**
Prompt:
You can try to generate the final answer for the [Question] by referring to the [Query]-[Answer] pairs, starting with [Final Content].
[Query 1]: "Who is the performer of Spirit If...?"
[Answer 1]: "Kevin Drew"
[Query 2]: "What is the place of birth of Kevin Drew?"
[Answer 2]: "Toronto"
[Query 3]:"Where do greyhound buses leave from in Toronto?"
[Answer 3]: "Toronto Coach Terminal "

Output:
The performer of Spirit If... is Kevin Drew [1]. Kevin Drew was born in Toronto [2]. Greyhound buses in Toronto leave from Toronto Coach Terminal [3]. So the final answer is Toronto Coach Terminal.

[1] ... Spirit If... is the debut solo album by Kevin Drew. It was released on September 18, 2007 ...
[2] ... Kevin Drew (born September 9, 1976 in Toronto) ...
[3] ...The Toronto Coach Terminal is the central bus station for inter-city services in Toronto, Ontario, Canada ... when it was leased out in its entirety to bus lines Coach Canada and Greyhound Canada ...

**Figure 21: Details (Round 3, Final).**

