# OpenReview forum: "Search-in-the-Chain: Interactively Enhancing Large Language Models with Search for Knowledge-intensive Tasks"
_ACM.org/TheWebConf/2024/Conference — TheWebConf24_

### Official Review · Reviewer_rAWs · 2023-11-20

**Novelty:** 5
**Technical Quality:** 5

**Review:**

This paper falls within the domain of retrieval-augmented generation for question answering, where responses from Large Language Models (LLMs) are augmented by Information Retrieval (IR) results. The paper specifically addresses the challenges posed by complex multi-hop questions involving long reasoning chains. The proposed approach operates as follows: when presented with a question, an LLM generates a query chain based on an example context. This query chain comprises a set of IR subqueries and their corresponding answers, with unresolved subquestions marked. An IR engine is then utilized to either verify the LLM-generated answers or provide information for the unresolved subquestions. Subsequently, the LLM is invoked again to regenerate a new query plan based on this additional information. This iterative process of query chain generation and IR retrievals continues for up to 5 steps. The final answer includes a trace of the query path along with references to supporting documents.

Through an iterative process involving interactions between the LLM and IR calls, the authors improve the performance of their question-answering system compared to one-shot solutions. The one-shot solutions used for comparison include those without IR interaction, such as Chain-of-Thought (CoT), Auto-CoT, and Least-to-Most, as well as those incorporating IR, such as Tree-of-Thought with IR and DSP.

The approach is sound and the performance improvement is good. The presentation is ok but not great (see next).

A few areas to improve the presentation:
 - There are too many references to "previous approaches" and what "they don't do" without knowing what the authors are doing and why. This becomes a bit irritating. Please cut the chase and tell what you are doing and not what previous  approaches don't.
 - What do we learn from this work? When do we expect this approach to be more or less effective?
 - Discuss the limitations of the work. Those multiple LLM calls will cost more compared to the related work. Other than that, when are the result worse?

**Questions:**

Q1- How many times is LLM called for query path generation or regeneration? Is it until all nodes in a path become green? How much sensitive is this to those thresholds?
Q2. Based on the concept of self-consistency [a], we know that one can improve the performance by simply calling LLM multiple times even with the input kept the same. How do we know what being observed here is not the result of self-consistency?
Q3. How do we know the approach is not tied to one specific LLM (in this case, GPT-3.5-turbo) and if the results are consisten in other LLMs? Have you tried any non-OpenAI models?
Q4. How good/fair is Cover-EM in comparing different models? As this metric checks if the ground truth is in the generated answer, a system that returns longer answers will have an advantage here. What was the length distribution of the answers across different models being compared?


[a] Wang, X., Wei, J., Schuurmans, D., Le, Q., Chi, E., Narang, S., Chowdhery, A. and Zhou, D., 2022. Self-consistency improves chain of thought reasoning in language models. arXiv preprint arXiv:2203.11171.

**Ethics Review Description:**

I don't see any issues.

**Reviewer Confidence:**

4: The reviewer is certain that the evaluation is correct and very familiar with the relevant literature

**Scope:**

4: The work is relevant to the Web and to the track, and is of broad interest to the community

---

### Official Review · Reviewer_svYf · 2023-11-20

**Novelty:** 5
**Technical Quality:** 5

**Review:**

The paper presents a novel framework, SearChain, designed to enhance the interaction between Large Language Models (LLMs) and Information Retrieval (IR) systems. The proposed SearChain framework introduces a new way to incorporate IR into LLMs, which improves the LLM's accuracy, credibility, and traceability. The experiments are comprehensive, and the results show that SearChain outperforms state-of-the-art baselines on complex knowledge-intensive tasks.

Pros:
- The paper addresses an important problem in the field.
- The proposed SearChain framework is novel and innovative.
- The work shows promising results in improving the performance of LLMs in complex tasks.

Cons:
- While the task is well motivated and challenges are well highlighted, some of the claims seem overhyped. For instance, in section 2.1, the claim that "...They focus on how to solve local sub-questions..." might not hold as chat-LLM will always remember the previous context.
- The chain-of-query generation is based on prompting LLMs. However, the paper lacks an evaluation of this task as a query can be segmented differently in terms of quantity, difficulty, and granularity.
- When there is a need for external knowledge, "[Unsolved query]" is tagged. This step also requires evaluation.
- It would be interesting to explore how the proposed framework performs with different types of IR systems.
- While it is great that this paper has compared against multiple existing methods as baselines, fair baselines should only be based on GPT-3.5-turbo as whatever prompts used, GPT-3.5 might have existing knowledge of the test sets.
- The tree-based approach makes sense. However, efficiency should also be tracked.
- The overall novelty is limited

**Questions:**

- Could you clarify the claim made in section 2.1 regarding LLMs focusing on how to solve local sub-questions? Given that chat-LLMs can remember the previous context, how does this affect your claim?
- The chain-of-query generation is based on prompting LLMs. Could you provide more information on how you evaluated this task, especially considering that a query can be segmented differently in terms of quantity, difficulty, and granularity?
- When external knowledge is needed, an "[Unsolved query]" is tagged. Could you elaborate on how you evaluated this step and its effectiveness?
Have you considered exploring how the proposed SearChain framework performs with different types of IR systems? If so, could you share some preliminary insights or plans for future work in this direction?
- The paper compares the proposed framework against multiple existing methods as baselines. However, could you comment on the fairness of these baselines, especially considering that whatever prompts are used, GPT-3.5 might have existing knowledge of the test sets?
- While the tree-based approach seems sensible, could you shed some light on the efficiency of this approach? Have you tracked the efficiency, and if so, could you share the results?
- Lastly, could you discuss any potential limitations of your approach that you haven't covered in the paper? How do you plan to address these in future work?

**Reviewer Confidence:**

3: The reviewer is confident but not certain that the evaluation is correct

**Scope:**

4: The work is relevant to the Web and to the track, and is of broad interest to the community

---

### Official Review · Reviewer_KMD7 · 2023-11-22

**Novelty:** 4
**Technical Quality:** 4

**Review:**

Pros：
1.The paper introduces a novel framework (SearChain) that enhances the credibility of answers generated by LLM by employing Information Retrieval (IR) for validation and correction. This effectively alleviates the issue of misinformation from IR errors misleading the LLM, providing valuable insights for future work.
2.The experimental design of this paper is comprehensive, outperforming the best baselines across multiple metrics.
3.The case study is highly detailed and specific, effectively illustrating the workflow of SearChain and addressing some uncertainties.
Cons:
The paper's innovation is somewhat limited, as the proposed novel retrieval paradigm still relies on the retrieval method from previous work (DSP).

**Questions:**

1.In comparing SearChain with two different kinds of baselines, particularly against the baseline introducing IR to LLM, is the retrieval method used by SearChain consistent with the other baselines?
2.How generalizable is the framework? Does it apply to other retrieval methods and LLMs, and does it demonstrate improvements in performance?

**Reviewer Confidence:**

3: The reviewer is confident but not certain that the evaluation is correct

**Scope:**

3: The work is somewhat relevant to the Web and to the track, and is of narrow interest to a sub-community

---

### Official Review · Reviewer_NaYz · 2023-11-23

**Novelty:** 6
**Technical Quality:** 6

**Review:**

SearChain innovatively combines LLMs with IR to tackle issues like compositional reasoning over multiple knowledge domains, memorization of vast and real-time knowledge, and avoiding factually inconsistent hallucinations. The framework operates at a 'chain' level as opposed to traditional methods that focus on individual nodes. It leverages a Chain-of-Query (CoQ) method, where LLMs generate a reasoning chain consisting of IR-oriented queries and corresponding answers. IR then verifies and corrects these answers, improving the overall credibility and accuracy of the LLMs. This process also ensures that LLMs receive only the knowledge they lack, minimizing misinformation risks. Additionally, SearChain enhances the traceability of generated content by marking references to supporting documents.

Strengths:
- Innovative integration of IR and LLM. SearChain's approach to combining IR with LLMs at the chain level is a significant advancement over traditional node-level methods, allowing for more comprehensive and coherent reasoning.
- Enhanced accuracy and credibility. By verifying and correcting each step of the reasoning chain, SearChain significantly improves the accuracy and credibility of the LLMs in handling complex tasks.
- Improved traceability. The framework's ability to mark references to supporting documents for each step of the reasoning process enhances the traceability and reliability of the content generated by LLMs.

Weaknesses:
- Increased complexity in the problem-solving process. While SearChain's chain-based method enhances the accuracy of LLM-generated results, it also elevates the complexity of executing problem-solving tasks. This increased complexity raises questions about the efficiency of this model compared to traditional LLMs with IR integration and LLMs without IR. A detailed analysis of the time overheads in these different scenarios would be beneficial to understand the trade-offs between accuracy and efficiency.

- Scenarios with inconclusive IR and LLM outputs. This paper highlights that SearChain can introduce high-reliability answers through verification. However, there might be situations where the LLM is unable to provide effective responses, and simultaneously, IR fails to retrieve reliable evidence. In such cases, it's unclear how SearChain would handle the absence of both a solid LLM output and conclusive IR support. Understanding the fallback mechanisms or alternative strategies in such scenarios would be crucial.

- Formatting issues.  Attention to detail such as uniformity in figures and tables presentation, and overall document layout can significantly enhance the readability and professional appearance of the paper.

**Questions:**

None

**Reviewer Confidence:**

3: The reviewer is confident but not certain that the evaluation is correct

**Scope:**

4: The work is relevant to the Web and to the track, and is of broad interest to the community

---

### Official Review · Reviewer_mny7 · 2023-11-25

**Novelty:** 6
**Technical Quality:** 6

**Review:**

In this paper, the authors propose a novel method to improve LLMs responses for knowledge-intensive tasks using search. The novelty is in the global planning of reasoning chain in a tree form and using IR to verify the nodes and explore in depth-first strategy.

Strengths:
1. Novel way to reason LLM response generation.
2. Extensive experiments with several datasets and baselines.
3. Well written and easy to follow.

Limitations:
1. Some related work is missing.
2. Choice of LLM is limited to a OpenAI model, none of the models with open weights are evaluated.
3. Error analysis of cases when SerChain would fail is needed.

**Questions:**

1. More detailed comparison to Tree of Thought is needed since it is similar in idea. How does breadth-first strategy perform compared to depth-first strategy?
2. Some relevant works like RARR [1] should be compared,  and more recent works like Chain of Verification [2] and [3] can be cited.
3. How would the tree-based exploration in SerChain perform for questions that need reasoning? for example, numerical QA using FinQA dataset. If the final answer requires dependency of several sub-questions and their answers, then tree like exploration may not be the best match.
4. It would be beneficial to see some analysis of failure cases for SerChain. Also what happens when the question is unanswerable
5. How would SerChain and the baselines perform on models with open weights like Llama and Vicuna etc? Also how well does it work when using smaller models?
6. While the efficiency analysis shows that the SerChain is not much more expensive than the baselines, it is limited setting with a fixed smaller corpus. When running this method on open Web or larger corpora like common crawl this method could be prohibitively expensive.
7. Are the results statistically significant?

[1] Gao, Luyu, Zhuyun Dai, Panupong Pasupat, Anthony Chen, Arun Tejasvi Chaganty, Yicheng Fan, Vincent Zhao et al. "Rarr: Researching and revising what language models say, using language models." In Proceedings of the 61st Annual Meeting of the Association for Computational Linguistics (Volume 1: Long Papers), pp. 16477-16508. 2023.

[2] Dhuliawala, Shehzaad, Mojtaba Komeili, Jing Xu, Roberta Raileanu, Xian Li, Asli Celikyilmaz, and Jason Weston. "Chain-of-verification reduces hallucination in large language models." arXiv preprint arXiv:2309.11495 (2023).

[3] Zheng, Chuanyang, Zhengying Liu, Enze Xie, Zhenguo Li, and Yu Li. "Progressive-hint prompting improves reasoning in large language models." arXiv preprint arXiv:2304.09797 (2023).

**Reviewer Confidence:**

4: The reviewer is certain that the evaluation is correct and very familiar with the relevant literature

**Scope:**

4: The work is relevant to the Web and to the track, and is of broad interest to the community

---

### Decision · Program_Chairs · 2024-01-22

**Decision:**

Accept

**Comment:**

This is the meta-review by the area chair responsible for your paper, taking into account the reviews and later discussion, as well as my own reading of the work.

 Synthesis: I won't repeat the comments from the reviewers, which are clear and self-explanatory. The reviewers express great interest in problem of using IR to ground generative LLM responses. While the paper doesn't resolve this highly complex problem in a definite way, it makes some solid first steps. We trust that the authors will look very carefully at the reviewers' suggestions, and do their best to incorporate them in the final camera ready copy.

 Final Disposition:

 The final outcome of the reviews and discussion is a recommendation to accept the paper.